# Speedrunning ImageNet Diffusion

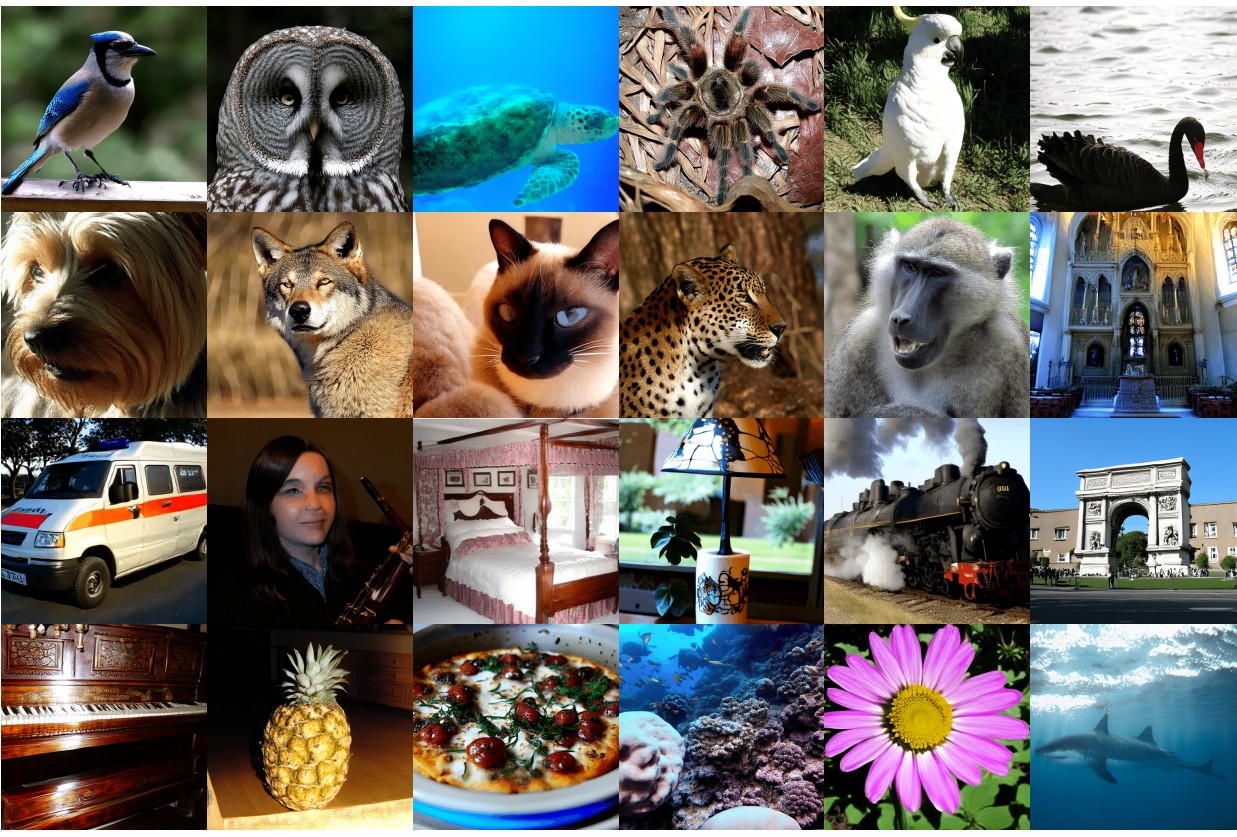

Figure 1: SR-DiT-B/1 samples on ImageNet-512.

## Abstract

Recent advances have significantly improved the training efficiency of diffusion transformers. However, these techniques have largely been studied in isolation, leaving unexplored the potential synergies from combining multiple approaches. We present **SR-DiT** (Speedrun Diffusion Transformer), a framework that systematically integrates token routing, architectural improvements, and training modifications on top of representation alignment. Our approach achieves FID 3.14 and KDD 0.290 on ImageNet-256 using only a 140M parameter model at 400K iterations without classifier-free guidance—comparable to results from 685M parameter models trained significantly longer. To our knowledge, this is a state-of-the-art result at this model size. Through extensive ablation studies, we identify which technique combinations are most effective and document both synergies and incompatibilities. We release our framework as a computationally accessible baseline for future research.

# 1 Introduction

Diffusion models have emerged as the dominant paradigm for high-quality image generation, yet their training remains computationally expensive. Recent years have witnessed a proliferation of techniques aimed at accelerating diffusion model training: improved architectures Zhang et al. (2024), representation alignment Yu et al. (2024); Wu et al. (2025), better tokenizers Leng et al. (2025), and various training modifications. However, these advances have largely been developed and evaluated in isolation, each claiming improvements over increasingly outdated baselines. This fragmented landscape leaves a critical question unanswered: *how do these techniques interact when combined, and what performance is achievable by systematically integrating them?*

The current state of research presents several challenges. First, most techniques are evaluated against vanilla diffusion transformers, ignoring the substantial gains from other concurrent work. Second, flagship results typically require large models (e.g., SiT-XL with 685M parameters) trained for millions of iterations, creating high barriers for academic researchers with limited compute. Third, the interactions between techniques—whether synergistic or redundant—remain poorly understood.

We address these challenges with **SR-DiT** (Speedrun Diffusion Transformer), a framework that systematically combines recent advances to maximize training efficiency. Our key insight is that many techniques target orthogonal aspects of the training process: representation alignment provides strong learning signals, token routing reduces computational redundancy and improves information flow, modern architectures improve optimization dynamics, and semantic tokenizers provide more learnable latent spaces. By carefully integrating these components, we achieve results that rival or exceed large-scale models while using only a fraction of the compute.

Concretely, starting from the SiT-B/1 architecture (130M parameters), we add improvements to finally achieve FID 3.14 on ImageNet-256 at 400K iterations without classifier-free guidance. For comparison, REG Wu et al. (2025)—which itself claims 63× convergence speedup over vanilla SiT—requires SiT-XL (685M parameters) to reach FID 3.4, while REPA Yu et al. (2024) needs 4 million training steps with SiT-XL to achieve FID 5.9. Our approach thus demonstrates that combining existing techniques intelligently can yield outsized gains, providing a strong and efficient baseline for future research.

Our contributions are:

- A systematic study of how recent diffusion training techniques interact when combined, identifying synergies and incompatibilities.

- An efficient framework achieving FID 3.14 and KDD 0.290 on ImageNet-256 with only 140M parameters at 400K iterations, comparable to much larger models trained for longer.

- Extensive ablations documenting both successful combinations and negative results, providing practical guidance for researchers.

- A computationally accessible baseline enabling faster iteration for academic research.

Because we draw together many techniques from across the recent literature, we summarize the abbreviations and method names used throughout the paper in Table 1 for ease of reference.

# 2 Related Work

**Diffusion models.** Denoising diffusion probabilistic models Ho et al. (2020); Sohl-Dickstein et al. (2015) and score-based generative models Song & Ermon (2019); Song et al. (2021) have become the foundation for state-of-the-art image generation. Flow matching Lipman et al. (2023); Liu et al. (2022) provides an alternative formulation with simpler training objectives. The Diffusion Transformer (DiT) Peebles & Xie (2023) demonstrated that transformer architectures can match or exceed U-Net performance, while SiT Ma et al. (2024) extended this to flow matching. We build upon SiT as our base architecture.

Table 1: Terminology and abbreviations used in this paper.

| Term | Meaning |
| --- | --- |
| *Architectures & methods* | |
| SR-DiT | Speedrun Diffusion Transformer (this work) |
| DiT | Diffusion Transformer Peebles & Xie (2023) |
| SiT | Scalable Interpolant Transformer (flow-matching DiT; our base) Ma et al. (2024) |
| REPA | Representation Alignment: align hidden states to a vision encoder Yu et al. (2024) |
| REG | Representation Entanglement for Generation: jointly diffuse the DINO CLS token Wu et al. (2025) |
| SPRINT | Sparse-dense residual-fusion token routing (75% drop) Zhang et al. (2024) |
| TREAD | Token Routing for Efficient Architecture-agnostic Diffusion Krause et al. (2025) |
| CFM | Contrastive Flow Matching Stoica et al. (2025) |
| Value Residual | Value Residual Learning: residual across the value stream Zhou et al. (2024) |
| RoPE | Rotary Position Embeddings Su et al. (2021) |
| RMSNorm | Root-Mean-Square Normalization Zhang & Sennrich (2019) |
| QK Norm | Query–Key Normalization Henry et al. (2020) |
| Muon | Momentum-orthogonalized optimizer Jordan (2024) |
| *Guidance & sampling* | |
| CFG | Classifier-Free Guidance |
| PDG | Path-Drop Guidance (qualitative samples only) |
| NFE | Number of Function Evaluations (sampling steps) |
| *Evaluation metrics* | |
| KDD | Kernel DINO Distance Stein et al. (2023) |
| FID | Fréchet Inception Distance Heusel et al. (2017) |
| sFID | Spatial FID Nash et al. (2021) |
| IS | Inception Score Salimans et al. (2016) |
| Prec./Rec. | Precision / Recall Kynkäänniemi et al. (2019) |
| *Encoders & tokenizers* | |
| DINOv2 | Frozen vision encoder providing representation targets Oquab et al. (2023) |
| SD-VAE | Stable Diffusion VAE, 8× compression Rombach et al. (2022) |
| INVAE | Semantic VAE from REPA-E (E2E-INVAE), 16× compression Leng et al. (2025) |

**Representation alignment.** REPA Yu et al. (2024) introduced the idea of aligning diffusion model hidden states with pretrained vision encoder features, achieving significant training speedups. REG Wu et al. (2025) extended this with generative objectives, claiming 63× convergence speedup over vanilla SiT. These methods provide strong learning signals that guide the model toward semantically meaningful representations. We use REG as our starting point and evaluate additional techniques on top.

**Semantic latent spaces.** The choice of image tokenizer significantly impacts diffusion training dynamics. Standard SD-VAE Rombach et al. (2022) encodes images into latent spaces optimized for reconstruction, but not necessarily for generation. LightningDiT Yao et al. (2025) introduced a VAE trained with semantic objectives, producing latent spaces that are more "diffusable"—easier for diffusion models to learn. IN-VAE Leng et al. (2025) (from REPA-E) follows this direction with improved semantic properties. These tokenizers accelerate learning by providing latent representations with stronger semantic structure.

**Token routing.** TREAD Krause et al. (2025) demonstrated that routing 50% of tokens to skip intermediate transformer layers both reduces computation and improves convergence—a counterintuitive finding suggesting that full token processing may be redundant. SPRINT Zhang et al. (2024) introduces architectural modifications that allow increasing the token drop rate to 75%, achieving greater efficiency gains. These methods reveal that diffusion transformers have significant computational slack that can be exploited for efficiency.

**Architectural improvements.** Modern transformer components from language modeling have shown benefits in vision. LightningDiT incorporates SwiGLU activations Shazeer (2020), RMSNorm Zhang & Sennrich (2019), and RoPE Su et al. (2021). QK normalization Henry et al. (2020) stabilizes attention, and

Value Residual Learning Zhou et al. (2024) improves information flow. We evaluate how these architectural choices interact with representation alignment and token routing.

## 3 Background

### 3.1 Evaluation Metrics

While Fréchet Inception Distance (FID) Heusel et al. (2017) has been the standard metric for evaluating generative models, recent work has exposed significant limitations in commonly used metrics Stein et al. (2023). Stein et al. demonstrate that among existing metrics, **Kernel DINO Distance (KDD)** correlates most strongly with human perceptual judgments. KDD computes distances between generated and real image distributions using DINOv2 Oquab et al. (2023) features in a kernel-based framework. Lower KDD values indicate better generation quality. We report both KDD and traditional metrics (FID, sFID, IS, Precision, Recall) for comprehensive evaluation.

### 3.2 Diffusion / Flow Matching

We follow SiT Ma et al. (2024) and train the model with a flow-matching objective using velocity prediction. Given a clean input $x$ (VAE latents) and a timestep $t \in [0, 1]$, we construct a noisy sample using an interpolant

$$x_t = \alpha(t)\, x + \sigma(t)\, \epsilon, \qquad \epsilon \sim \mathcal{N}(0, I), \tag{1}$$

where $\alpha(t), \sigma(t)$ define the path (e.g., linear or cosine). The corresponding velocity target is

$$v_t^\star = \dot{\alpha}(t)\, x + \dot{\sigma}(t)\, \epsilon. \tag{2}$$

The base training loss is mean-squared error on velocity prediction, $\mathbb{E}[\|v_\theta(x_t, t) - v_t^\star\|_2^2]$.

### 3.3 Time Shifting

We use time shifting Esser et al. (2024) to reweight which timesteps the model sees, reducing over-emphasis on high-SNR (easy) denoising steps. We first sample $t \sim \mathcal{U}(0, 1)$, then apply the monotone transform

$$t' = \frac{s\, t}{1 + (s-1)t}, \qquad s = \sqrt{\frac{D}{4096}}, \qquad D = C \cdot H \cdot W, \tag{3}$$

which (for $s > 1$) shifts mass toward larger $t$ (noisier / lower-SNR inputs). We apply the same transformation during both training and sampling. Following Zheng et al. Zheng et al. (2024), we compute the shift factor from the full latent dimensionality (including channels) and use 4096 as the reference dimension.

### 3.4 Rotary Position Embeddings (RoPE)

Rotary position embeddings (RoPE) Su et al. (2021) encode positional information by rotating query and key vectors in multi-head attention. For a token at position $i$ and head vector $q_i$ (and similarly $k_i$), RoPE rotates each consecutive 2D pair via

$$\text{RoPE}(q_i) = q_i \odot \cos \omega_i + \text{rot}(q_i) \odot \sin \omega_i, \quad \text{rot}([x_{2j}, x_{2j+1}]) = [-x_{2j+1},\, x_{2j}], \tag{4}$$

where $\omega_i$ are position-dependent frequencies and $\odot$ is elementwise multiplication. For images, we use a 2D extension where positions $i$ correspond to indices on the flattened $H \times W$ patch grid.

### 3.5 RMSNorm

RMSNorm Zhang & Sennrich (2019) is a normalization layer that scales activations by their root-mean-square (without mean-centering), which reduces computation and can improve stability. For an input vector $x \in \mathbb{R}^d$, RMSNorm computes

$$\text{RMSNorm}(x) = g \odot \frac{x}{\sqrt{\frac{1}{d} \sum_{j=1}^{d} x_j^2 + \epsilon}}, \tag{5}$$

where $g$ is a learned gain parameter and $\epsilon$ is a small constant.

### 3.6 Value Residual Learning

Value Residual Learning Zhou et al. (2024) modifies attention by injecting a residual connection across the *value* stream. The method caches the value vectors from an early attention block and reuses them as a reference value for subsequent blocks. Let $v^{(\ell)}$ denote the value vectors produced by block $\ell$ (after the value projection) and let $v^{(0)}$ denote the cached reference values (from the first block). We form a mixed value

$$\tilde{v}^{(\ell)} = \lambda\, v^{(0)} + (1 - \lambda)\, v^{(\ell)}, \tag{6}$$

with a learned scalar $\lambda \in [0, 1]$ (implemented as a learnable parameter). Attention then uses $\tilde{v}^{(\ell)}$ in place of $v^{(\ell)}$.

### 3.7 Token Routing and Path-Drop Guidance

TREAD Krause et al. (2025) and SPRINT Zhang et al. (2024) implement token routing by temporarily dropping a large fraction of tokens in the middle transformer blocks: a dense prefix processes all tokens, then only the retained (sparse) tokens are propagated through a sequence of mid blocks, and the dropped tokens are reintroduced for the final blocks. SPRINT improves this reintroduction step by padding the sparse sequence back to the full length with a learned `[MASK]` token and explicitly fusing the padded sparse stream with the dense stream (e.g., concatenation followed by a projection).

We also use *path-drop guidance* (PDG), a CFG-style heuristic in which the unconditional prediction is intentionally weakened by skipping the routed mid blocks entirely (i.e., dropping the sparse path). Let $v_\theta(x_t, t, c)$ denote the conditional model prediction under class conditioning $c$ (sparse path enabled) and $v_\theta^{\text{weak}}(x_t, t)$ the unconditional prediction computed with the routed mid blocks skipped. The guided prediction is

$$v_\theta^{\text{guide}}(x_t, t, c) = v_\theta^{\text{weak}}(x_t, t) + s\left(v_\theta(x_t, t, c) - v_\theta^{\text{weak}}(x_t, t)\right), \tag{7}$$

where $s$ is the guidance scale. Importantly, we use PDG **only for qualitative sampling** (visualizations) and **do not** use PDG for the quantitative metrics reported in the paper.

### 3.8 Contrastive Flow Matching

Contrastive Flow Matching (CFM) Stoica et al. (2025) introduces an additional training objective that improves convergence speed. The CFM loss is computed by contrasting model outputs with random targets:

$$\mathcal{L}_{\text{CFM}} = -\lambda\mathbb{E}\left[\left\|v_\theta(x_t, t) - \hat{v}_{\text{target}}\right\|^2\right] \tag{8}$$

where $\hat{v}_{\text{target}}$ is a random training target unrelated to $x_t$ (obtained by shuffling the minibatch of the velocity target elementwise), and $\lambda$ is a weighting coefficient (default 0.05). The negative sign encourages the model to maximize distance from other samples' predictions.

### 3.9 Representation Alignment (REPA and REG)

We build on Representation Alignment for Generation (REPA) Yu et al. (2024) and its extension REG Wu et al. (2025), which add an auxiliary representation objective to the standard denoising / flow-matching loss. We use the same noising process and velocity-prediction objective described above.

For the representation targets, we pass the *clean* image through a frozen vision encoder (DINOv2 Oquab et al. (2023)) to obtain per-token targets $z$ (and a global CLS embedding $c_{\text{cls}}$). The diffusion transformer produces intermediate hidden states that are mapped through small MLP projectors to predicted representations $\tilde{z}$. REPA uses a projection loss based on cosine similarity:

$$\mathcal{L}_{\text{REPA}} = -\lambda_{\text{REPA}}\,\frac{1}{M}\sum_{m=1}^{M}\left\langle \frac{z_m}{\|z_m\|_2},\, \frac{\tilde{z}_m}{\|\tilde{z}_m\|_2}\right\rangle, \tag{9}$$

where $m$ indexes tokens and we use $\lambda_{\mathrm{REPA}} = 0.5$ by default.

REG extends this setup by also diffusing the DINO CLS embedding alongside the latents.

# 4 Method

Our approach systematically combines recent advances to maximize training efficiency. We start from REG Wu et al. (2025) with INVAE Leng et al. (2025) as our baseline, then progressively add architectural improvements and training objective modifications.

## 4.1 Architectural Improvements

We evaluate several modern transformer components that have shown benefits in language and vision models:

**SPRINT** Zhang et al. (2024): We use SPRINT token routing with a drop ratio of 0.75 (keeping 25% tokens) in the sparse path. Following the standard SPRINT split, we run 2 dense "encoder" blocks on all tokens, route over the middle blocks (operating on the sparse subset, then padding back and fusing), and run 2 dense "decoder" blocks on the fused full sequence.

**RMSNorm** Zhang & Sennrich (2019): Starting from our fork of the REG baseline, we replace all LayerNorm instances in the backbone with RMSNorm. Concretely, this includes: (i) the two per-block normalizations before attention and before the MLP in every transformer block (`norm1`, `norm2`); (ii) the per-head query/key normalizers used by QK Norm (`q_norm`, `k_norm`); (iii) the final normalization before the output projection (`norm_final`); and (iv) the RMSNorm applied to the REG-diffused CLS embedding before concatenation (`wg_norm`).

**Rotary Position Embeddings (RoPE)** Su et al. (2021): We use 2D RoPE (see Section 3.4) via the EVA-02-style implementation Fang et al. (2023), later adapted by the TREAD authors to support routed / sparse token subsets for their LightningDiT+TREAD experiments Yao et al. (2025); Krause et al. (2025).[1] In routed blocks, we pass per-token indices `rope_ids` so each retained token is rotated using its original spatial position. We further modify the implementation to exclude the leading class token (the REG-diffused CLS embedding) from rotation.

**QK Normalization** Henry et al. (2020): Normalizing query and key vectors before computing attention scores stabilizes training dynamics.

**Value Residual Learning** Zhou et al. (2024): Adding a residual connection around the value projection improves gradient flow and model expressiveness.

**Layerwise Scaling**: We vary the MLP (feedforward) hidden width across depth—wider in the deeper transformer blocks and narrower in the earlier ones—while holding the total parameter count fixed. We keep it in the final model because it lowers sFID at a marginal (0.01) cost in FID (Table 2).

These components were proposed independently in various contexts; our contribution is evaluating their interactions when combined with representation alignment and token routing.

## 4.2 Training Objective

We incorporate Contrastive Flow Matching (CFM) Stoica et al. (2025), which adds an auxiliary loss that improves convergence speed (see Section 3). Overall, our training loss is

$$\mathcal{L} = \mathcal{L}_{\mathrm{vel}} + \lambda_{\mathrm{REPA}}\,\mathcal{L}_{\mathrm{REPA}} + \lambda_{\mathrm{cls}}\,\mathcal{L}_{\mathrm{cls}} + \lambda_{\mathrm{CFM}}\,\mathcal{L}_{\mathrm{CFM}}, \tag{10}$$

where $\mathcal{L}_{\mathrm{vel}}$ is the standard velocity-prediction MSE on latents (Section 3), $\mathcal{L}_{\mathrm{REPA}}$ is the projection loss (Section 3), $\mathcal{L}_{\mathrm{cls}}$ is the velocity-prediction MSE for the REG-diffused CLS embedding, and $\mathcal{L}_{\mathrm{CFM}}$ is the contrastive term. We use $\lambda_{\mathrm{REPA}} = 0.5$, $\lambda_{\mathrm{cls}} = 0.03$, and $\lambda_{\mathrm{CFM}} = 0.05$.

---

[1] https://github.com/flixmk/LightningDiT_TREAD/blob/main/models/pos_embed_tread.py

### 4.3 Time Shifting

We apply time shifting (Section 3.3) during both training and sampling.

## 5 Experiments

### 5.1 Experimental Setup

We conduct experiments on ImageNet-256 Deng et al. (2009). We build upon the SiT-B Ma et al. (2024) and REG Wu et al. (2025) architecture and compare our ablations against multiple baselines. Our model uses the SiT-B/1 architecture (patch size 1 instead of 2), starting with 132M parameters for the baseline, as INVAE has $16\times$ spatial compression compared to SD-VAE Rombach et al. (2022)'s $8\times$ compression. SPRINT modifications increase the model to 140M parameters. This architecture choice significantly reduces computational costs compared to larger models like SiT-XL (685M parameters) while maintaining strong performance.

Training the final SR-DiT-B/1 architecture to 400K iterations took approximately 10 hours on a single node with $8\times$ NVIDIA H200 GPUs for ImageNet-256 (80 GPU-hours), and approximately 15 hours for ImageNet-512 (120 GPU-hours).

We evaluate generation quality using Kernel DINO Distance (KDD) alongside standard metrics: Fréchet Inception Distance (FID) Heusel et al. (2017), spatial FID (sFID) Nash et al. (2021), Inception Score (IS) Salimans et al. (2016), Precision, and Recall Kynkäänniemi et al. (2019). All metrics are computed on 50K generated samples. Following RAE Zheng et al. (2024), we use balanced label sampling during generation for metric calculation, ensuring each class is equally represented in the generated samples. This corrects an implementation detail where random sampling can lead to imbalanced class distributions, improving FID marginally. **All results are reported without CFG or PDG**.

### 5.2 Results

We use REG with E2E-INVAE (henceforth referred to as INVAE) as our base configuration and systematically evaluate architectural and training improvements. Table 2 presents our main results at 400K iterations, comparing against SiT-B/2 baselines.

In addition to ImageNet-256, we evaluate SR-DiT-B/1 on ImageNet-512 using the same training setup scaled to $512 \times 512$ resolution. We compare against the DiT-XL/2 and U-DiT-B baselines reported in U-DiTs Tian et al. (2024) as these are the only reported results we could find for FID-50k on ImageNet-512 at 400k iterations. As shown in Table 3, SR-DiT-B/1 achieves strong ImageNet-512 performance. All results are evaluated without CFG or PDG.

We caution that this ImageNet-512 comparison is limited: most recent efficient diffusion transformers report 512 numbers only with classifier-free guidance, much longer training, or larger models, so few results are directly comparable to our no-guidance, 400K, B-sized setting and the baseline set is small (and SR-DiT-B/1 is in fact smaller than the DiT-XL/2 baseline it surpasses). The 512 results should therefore be read as evidence that the recipe transfers to higher resolution rather than as an exhaustive state-of-the-art benchmark.

Figure 3 qualitatively illustrates how our final architecture sharpens details and improves semantic fidelity over the REG + INVAE baseline under identical sampling conditions.

**Baseline comparison.** The original SiT-B/2 Ma et al. (2024) with 130M parameters requires 400K iterations to achieve FID 33.0. REPA Yu et al. (2024) improves this to FID 24.4, while REG Wu et al. (2025) reaches FID 15.2 at 400K iterations with 132M parameters. Our baseline configuration (REG + INVAE) with SiT-B/1 achieves FID 10.56 using 132M parameters, already surpassing REG with SD-VAE at the same parameter count.

Table 2: Performance comparison on ImageNet-256 at 400K iterations. Lower FID/sFID/KDD and higher IS/Precision/Recall are better. All methods evaluated at NFE=250 without CFG or PDG.

| Config | Method | #Params | FID↓ | sFID↓ | IS↑ | Prec.↑ | Rec.↑ | KDD↓ |
|---|---|---|---|---|---|---|---|---|
| *Baselines (SiT-B/2, SD-VAE)* | | | | | | | | |
| – | SiT-B/2 | 130M | 33.0 | 6.46 | 43.7 | 0.53 | 0.63 | – |
| – | + REPA | 130M | 24.4 | 6.40 | 59.9 | 0.59 | 0.65 | – |
| – | + REG | 132M | 15.2 | 6.69 | 94.6 | 0.64 | 0.63 | – |
| *SR-DiT (SiT-B/1, INVAE)* | | | | | | | | |
| $SR\text{-}DiT_1$ | REG + INVAE | 132M | 10.56 | 5.49 | 104.5 | 0.691 | **0.632** | 0.586 |
| $SR\text{-}DiT_2$ | + SPRINT | 140M | 4.58 | 7.04 | 188.7 | 0.760 | 0.562 | 0.385 |
| $SR\text{-}DiT_3$ | + RMSNorm | 140M | 4.56 | 6.58 | 186.8 | 0.772 | 0.562 | 0.381 |
| $SR\text{-}DiT_4$ | + RoPE | 140M | 4.09 | 6.22 | 194.2 | 0.778 | 0.563 | 0.368 |
| $SR\text{-}DiT_5$ | + QK Norm | 140M | 4.02 | 6.18 | 193.5 | 0.784 | 0.563 | 0.364 |
| $SR\text{-}DiT_6$ | + Value Residual | 140M | 3.64 | 5.98 | 202.0 | 0.798 | 0.560 | 0.353 |
| $SR\text{-}DiT_7$ | + CFM | 140M | 3.61 | **5.26** | 211.9 | _0.817_ | 0.536 | 0.319 |
| $SR\text{-}DiT_8$ | + Time Shifting | 140M | 3.56 | 5.64 | 209.6 | 0.807 | 0.550 | 0.336 |
| $SR\text{-}DiT_9$ | + Balanced Sampling | 140M | 3.49 | 5.67 | _221.2_ | 0.808 | 0.546 | 0.332 |
| $SR\text{-}DiT_{10}$ | + Multi-Layer-REPA | 140M | 3.30 | _5.40_ | **239.2** | **0.820** | 0.548 | 0.298 |
| $SR\text{-}DiT_{11}$ | + Muon | 140M | **3.13** | 5.48 | 225.6 | 0.805 | _0.566_ | _0.291_ |
| $SR\text{-}DiT_{12}$ | + Layerwise Scaling | 140M | _3.14_ | _5.40_ | 225.8 | 0.807 | 0.565 | **0.290** |

Table 3: Performance comparison on ImageNet-512 at 400K iterations. Baseline results for DiT-XL/2* and U-DiT-B are taken from U-DiTs Tian et al. (2024). Lower FID/sFID/KDD and higher IS/Precision/Recall are better.

| Method | FID↓ | sFID↓ | IS↑ | Prec.↑ | Rec.↑ | KDD↓ |
|---|---|---|---|---|---|---|
| DiT-XL/2* | 20.94 | 6.78 | 66.30 | 0.745 | 0.581 | – |
| U-DiT-B | 15.39 | 6.86 | 92.73 | 0.756 | **0.605** | – |
| SR-DiT-B/1 (ours) | **4.23** | **5.46** | **192.34** | **0.839** | 0.531 | **0.306** |

## 5.3 Analysis

**What matters most.** The dominant improvement comes from representation alignment and entanglement (REPA/REG), token routing (SPRINT), and from using a semantic VAE (INVAE). These three produce the largest single jumps in Table 2: REG + INVAE reaches FID 10.56, and adding SPRINT alone more than halves it to 4.58.

**Other components help cumulatively.** RMSNorm, RoPE, QK Norm, and Value Residual are individually modest but consistently beneficial in our setting, improving optimization stability and information flow when stacked.

**A taxonomy of training bottlenecks.** It helps to organize the techniques by the bottleneck each primarily targets (Table 4). This grouping is a descriptive lens, not a tested claim that the categories are independent; with that caveat, it summarizes our results compactly: the largest reductions come from categories the baseline left unaddressed—semantic learning (REPA/REG, INVAE) and token routing—while the architecture, optimization, and loss/sampling refinements each add smaller cumulative gains.

**Synergy vs. redundancy.** The same lens fits the negatives (Appendix B): several discarded techniques target a bottleneck already covered and so add little—dispersive loss and SARA both overlap REG's representation objective, and Time-Weighted CFM overlaps standard CFM. We read this as a post-hoc pattern, not a predictive rule.

**Quality–diversity trade-off.** Several gains trade recall for precision: across the progression precision rises $0.691 \rightarrow 0.807$ while recall dips from 0.632 to a low of 0.536 (largest at SPRINT and CFM, whose contrastive

**Convergence Comparison on ImageNet-256**

Figure 2: Training convergence comparison on ImageNet-256. SR-DiT-B/1 achieves strong performance with substantial convergence speedup.

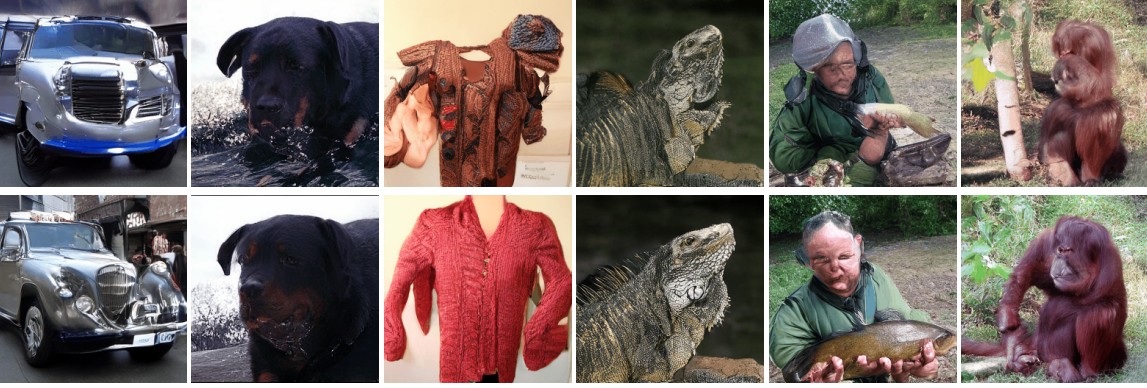

Figure 3: Qualitative comparison between our REG + INVAE starting point (top row) and the final SR-DiT-B/1 architecture (bottom row) on ImageNet-256 at 400K training iterations. Generated without CFG or PDG, using the same random seed and class label for both models.

term deliberately sharpens samples) before recovering to 0.565. This is a modest but real diversity cost; the distributional KDD metric—a kernel distance between the generated and real feature distributions, hence sensitive to coverage as well as quality—nonetheless falls overall, from 0.586 to 0.290. For applications that prioritize diversity over peak FID or efficiency, SPRINT and CFM are the natural candidates to drop, trading fidelity and efficiency back for coverage.

**On the ordering of components.** We report a single progressive path. The *final* configuration is order-invariant, but the per-component deltas are not—a component's measured value depends on what is already present, so we read them as path-dependent attributions rather than context-free contributions. $RELU^2$

Table 4: Taxonomy of the integrated techniques by the bottleneck each primarily addresses. We report the *relative* FID reduction each axis contributes along the Table 2 progression (one minus the product of post-step/pre-step FID ratios over that axis's steps), since a fixed absolute-FID drop is far more significant at low FID than at high FID. These reductions *compound* rather than sum: together they take FID from 33.0 to 3.14, a 90.5% reduction overall. The semantic axis includes the REPA/REG baseline gains and the SiT-B/1 + INVAE base change; all attributions are path-dependent (see "On the ordering of components").

| Bottleneck | Techniques | FID reduction |
|---|---|---|
| Semantic learning signal | REPA/REG, INVAE, Multi-Layer REPA | 70% |
| Computational redundancy & info. flow | SPRINT token routing | 57% |
| Expressivity & stability | RMSNorm, RoPE, QK Norm, | 20% |
| | Value Residual, Layerwise Scaling | |
| Optimization | Muon | 5% |
| Loss & sampling shaping | CFM, Time Shifting, Balanced Sampling | 4% |

makes this concrete (Appendix B): swapping $\text{GELU} \to \text{RELU}^2$ *improves* FID on its own ($4.02 \to 3.81$) but *worsens* it when Value Residual is already present ($3.64 \to 3.81$). The same change helps or hurts depending only on order, because the two are antagonistic—Value Residual's 0.38-FID gain on GELU vanishes on $\text{RELU}^2$. Mapping every such interaction would be $\mathcal{O}(n^2)$ rather than $\mathcal{O}(n)$ and is infeasible here; we flag residual order-sensitivity as a limitation (Section 6).

# 6 Discussion

## 6.1 Limitations and Future Work

**Scope of the efficiency claim.** Our efficiency comparison is against larger models that lack our recipe, so it shows the recipe's value at a fixed compute budget but does not isolate the effect of model size from that of the recipe, training budget, and their interaction; a controlled scaling study (the same recipe across SiT-B/L/XL) is the natural way to disentangle these and is left to future work.

**Generality across settings.** Our evaluation is confined to class-conditional ImageNet. Within this scope we provide some evidence that the gains are not an artifact of a single operating point: the recipe transfers from $256 \times 256$ to $512 \times 512$ (Table 3), and the improvements hold across the full training trajectory rather than only at 400K (Table 5). Nonetheless, validating the recipe on additional datasets and on more challenging settings remains important future work.

**Ordering and composition of components.** As discussed in Section 5.3, our per-component deltas are path-dependent attributions rather than context-free values; a full composition analysis (quadratic rather than linear in the number of components) is left to future work.

**Text to image generation.** Extending these techniques to text-to-image generation would demonstrate generalization and practical utility.

**Further optimizations.** There are numerous other techniques which we did not explore due to a lack of time and resources. Future work can investigate these techniques to further improve performance.

# 7 Conclusion

We present SR-DiT, a framework that achieves efficient diffusion model training by combining representation alignment with modern architectural improvements and training modifications. Starting from REG with INVAE (132M parameters), we demonstrate that progressive modifications (SPRINT, RMSNorm, RoPE, QK Norm, Value Residual, CFM, Time Shifting, Balanced Sampling, Multi-Layer REPA, Muon, Layerwise Scaling) create strong synergies, achieving FID 3.14 and KDD 0.290 on ImageNet-256 with the efficient SiT-B/1 architecture (140M parameters). We also adopt KDD as a more reliable evaluation metric. Our

systematic ablations identify which components contribute most to performance gains, providing insights for future work in efficient diffusion model training.

## Broader Impact Statement

Our work improves the training efficiency of diffusion transformers. Like most progress on generative image models, this efficiency is dual-use. On the beneficial side, lowering the compute required to reach strong generation quality broadens access for academic groups and individuals who lack industrial-scale resources, supports reproducible research, and reduces the energy footprint of training. The same property, however, lowers the barrier for harmful uses: more efficient training makes it cheaper for malicious actors to produce disinformation, non-consensual or impersonating imagery, and other deceptive synthetic media, and to do so at larger scale. We believe the efficiency techniques studied here are best understood as general-purpose accelerators that amplify whatever objective they are applied to, and should be deployed with that understanding.

**Released artifacts.** The checkpoints we release are *class-conditional* ImageNet-256/512 generators: they synthesize images from the 1000 fixed ImageNet object categories and are not conditioned on free-form text, faces, or individuals, which substantially limits their direct potential for targeted misuse relative to open-domain text-to-image systems. The *recipe* we describe is general, however, and could in principle be applied to more capable or differently conditioned models. We encourage practitioners who scale these techniques to more powerful or open-domain settings to pair them with standard safeguards—content provenance and watermarking, dataset documentation, output filtering, and gated or staged release where appropriate.

**Benchmark bias and evaluation.** ImageNet is known to carry demographic, geographic, and category-level biases, and the metrics we report inherit biases from their feature extractors (Inception for FID/IS, DINOv2 for KDD). Improvements on these aggregate metrics therefore do not guarantee fairness, balanced coverage, or the absence of biased or stereotyped generations, and should not be read as such. Our adoption of KDD is motivated by its stronger correlation with human perceptual judgment Stein et al. (2023), but it remains a distributional summary rather than a measure of social impact. Responsible deployment of any model trained with this recipe should include task-appropriate auditing beyond the metrics used here.

## Resources

Code, checkpoints, and experiment logs will be made publicly available upon acceptance.

### Acknowledgments

We gratefully acknowledge support from an industry partner for sponsoring the compute resources used in this work.

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

# A  Additional Results

This appendix documents additional quantitative results that complement the main text.

## A.1  Intermediate SR-DiT-B/1 Training Checkpoints

Table 5 reports SR-DiT-B/1 performance at intermediate training checkpoints, along with the baseline diffusion transformers and the final SR-DiT-B/1 model at 400K iterations.

Table 5: SR-DiT performance at intermediate training checkpoints on ImageNet-256. Lower FID/sFID/KDD and higher IS/Precision/Recall are better. All methods evaluated at NFE=250 without CFG or PDG.

| Method | Iter. | FID↓ | sFID↓ | IS↑ | Prec.↑ | Rec.↑ | KDD↓ |
|---|---|---|---|---|---|---|---|
| SiT-B/2 | 400K | 33.0 | 6.46 | 43.7 | 0.530 | 0.630 | – |
| + REPA | 400K | 24.4 | 6.40 | 59.9 | 0.590 | **0.650** | – |
| + REG | 400K | 15.2 | 6.69 | 94.6 | 0.640 | 0.630 | – |
| SR-DiT-B/1 | 50K | 21.55 | 7.75 | 61.37 | 0.677 | 0.475 | 0.926 |
| SR-DiT-B/1 | 100K | 9.17 | 6.52 | 124.69 | 0.767 | 0.484 | 0.584 |
| SR-DiT-B/1 | 200K | 4.91 | 5.95 | 184.05 | 0.801 | 0.523 | 0.408 |
| SR-DiT-B/1 (final) | 400K | **3.14** | **5.40** | **225.8** | **0.807** | **0.565** | **0.290** |

# B  Additional Ablations

This section documents ablation experiments that did not yield improvements, providing insights into which techniques are less effective when combined with representation alignment.

## B.1  Multi-Layer REPA Alignment

By default, we align the DINOv2-B features (layer 12 output) to the 2nd transformer block of SR-DiT$_9$ (SiT-B/1, INVAE). We additionally explore aligning different DINO layers and aligning multiple SR-DiT blocks simultaneously. Table 6 summarizes these ablations.

Aligning the 2nd, 4th, and 6th layers of our model to the DINO features yielded the best tradeoff for the results, and meaningfully improves over the default settings, so we adopt this configuration for SR-DiT$_{10}$.

Table 6: Multi-layer REPA alignment ablations for SR-DiT$_9$ at 400K iterations on ImageNet-256. Lower FID/sFID/KDD and higher IS/Precision/Recall are better. All results are evaluated at NFE=250 without CFG or PDG.

| Alignment (SR-DiT ← DINO) | Coeff | FID↓ | sFID↓ | IS↑ | Prec.↑ | Rec.↑ | KDD↓ |
|---|---|---|---|---|---|---|---|
| 2 ← 12 (default, SR-DiT$_9$) | 0.5 | 3.49 | 5.67 | 221.2 | 0.808 | 0.546 | 0.332 |
| 4 ← 12 | 0.5 | 3.62 | 5.51 | **244.7** | 0.832 | 0.525 | **0.288** |
| 2 ← 9 | 0.5 | 3.80 | **5.27** | 237.1 | **0.838** | 0.513 | 0.289 |
| 2 ← 9, 4 ← 12 | 0.5 | 3.35 | 5.55 | 243.4 | 0.823 | 0.542 | 0.296 |
| 2,4 ← 12 | 0.5 | **3.29** | 5.57 | 231.6 | 0.815 | 0.547 | 0.308 |
| 2,4,6 ← 12 (SR-DiT$_{10}$) | 0.5 | 3.30 | 5.40 | 239.4 | 0.820 | **0.549** | 0.298 |
| 2,4,6 ← 12 | 0.25 | 3.34 | 5.47 | 233.6 | 0.820 | **0.549** | 0.303 |
| 1,2,3,4,5,6 ← 12 | 0.5 | 3.40 | 5.40 | 235.1 | 0.822 | 0.544 | 0.300 |

## B.2  Alternative Activations

We evaluated replacing standard feedforward activations with SwiGLU Shazeer (2020), as proposed in LightningDiT. Table 7 shows results at 400K iterations.

Table 7: SwiGLU activation ablation at 400K iterations. Both experiments use REG + INVAE + SPRINT (140M parameters). SwiGLU shows marginal improvement with noticeable training slowdown.

| Method | FID↓ | sFID↓ | IS↑ | Prec.↑ | Rec.↑ | KDD↓ |
|---|---|---|---|---|---|---|
| SPRINT (baseline) | **4.58** | 7.04 | **188.7** | **0.760** | 0.562 | **0.385** |
| + SwiGLU | 4.65 | **6.84** | 185.1 | 0.759 | **0.568** | **0.385** |

SwiGLU showed marginal differences in performance metrics. While sFID improved slightly (6.84 vs 7.04) and KDD remained identical (0.385), FID slightly degraded (4.65 vs 4.58) and IS decreased (185.1 vs 188.7). Given these mixed results and a noticeable slowdown in training speed, we excluded SwiGLU from our final framework. The minimal performance differences did not justify the computational overhead.

We also tested additional activation variants. Table 8 shows results at 400K iterations.

Table 8: Alternative activation ablations at 400K iterations. All experiments use REG + INVAE + SPRINT + RMSNorm + RoPE + QK Norm. Standard GELU achieves the best compatibility with Value Residual.

| Activation | FID↓ | sFID↓ | IS↑ | Prec.↑ | Rec.↑ | KDD↓ |
|---|---|---|---|---|---|---|
| GELU (baseline) | 4.02 | **6.18** | 193.5 | **0.784** | 0.563 | 0.364 |
| RELU$^2$ | **3.81** | 6.47 | **198.4** | 0.778 | **0.572** | **0.351** |
| XieLU | 4.10 | 6.35 | 191.4 | 0.783 | 0.562 | 0.357 |
| Lopsided Leaky RELU$^2$ | 4.11 | 6.67 | 189.4 | 0.768 | 0.569 | 0.366 |

**RELU$^2$ incompatibility with Value Residual.** We tested RELU$^2$ So et al. (2021), which initially showed strong results (FID 3.81, KDD 0.351). However, when combined with Value Residual Learning, performance degraded significantly (FID 3.81 → 3.81, IS 198.4 → 195.1, KDD 0.351 → 0.355). In contrast, standard GELU with Value Residual achieved superior results (FID 3.64, IS 202.0, KDD 0.353). Since Value Residual provides larger gains than RELU$^2$ alone, we adopted GELU as our activation function. This is an antagonistic interaction rather than simple overlap: Value Residual's gain on GELU does not transfer to RELU$^2$, leaving RELU$^2$ with Value Residual (3.81) worse than GELU with Value Residual (3.64).

**XieLU.** We tested XieLU Huang & Schlag (2024), a recently proposed activation function derived through integration principles. While XieLU achieved competitive results (FID 4.10, KDD 0.357), it underperformed RELU$^2$ across most metrics and exhibited noticeably slower training speed. The performance gap and computational overhead made it unsuitable for our framework.

**Lopsided Leaky RELU$^2$.** We tested a variant of RELU$^2$ with negative slope handling, defined as:

$$f(x) = \begin{cases} x^2 & \text{if } x > 0 \\ \alpha x & \text{if } x \leq 0 \end{cases} \tag{11}$$

where we use $\alpha = 0.01$. While this variant showed marginally better training loss (approximately 0.0005 lower) than standard RELU$^2$, evaluation metrics were noticeably worse across all measures (FID 4.11 vs 3.81, KDD 0.366 vs 0.351). This discrepancy indicates the variant is prone to overfitting. We document this negative result because multiple researchers have independently experimented with this exact variation without success, yet no published work references it. By publishing this result, we hope to prevent others from expending computational resources on this unpromising direction.

### B.3 Alternative Normalization

We evaluated Derf Chen et al. (2025b), a recently proposed replacement for LayerNorm/RMSNorm that enables stronger normalization-free transformers. Table 9 shows results comparing RMSNorm (our baseline) with Derf on SR-DiT$_{11}$.

Derf substantially degraded FID (3.13 → 3.46), sFID (5.48 → 5.79), IS (225.6 → 198.8), precision (0.805 → 0.785), and KDD (0.291 → 0.317). While recall improved (0.566 → 0.584), the degradation across all

Table 9: Derf normalization ablation at 400K iterations. Replacing RMSNorm with Derf degrades performance across all metrics.

| Method | FID↓ | sFID↓ | IS↑ | Prec.↑ | Rec.↑ | KDD↓ |
|---|---|---|---|---|---|---|
| RMSNorm (SR-DiT$_{11}$) | **3.13** | **5.48** | **225.6** | **0.805** | 0.566 | **0.291** |
| Derf | 3.46 | 5.79 | 198.8 | 0.785 | **0.584** | 0.317 |

other metrics indicates that Derf is not suitable for our architecture. We retain RMSNorm in our final configuration.

### B.4   Dispersive Loss

The dispersive loss Wang & He (2025) for improving representation diversity yielded negligible performance differences compared to our baseline, confirming that representation alignment from REG already provides sufficient diversity in the learned representations.

### B.5   SARA Structural Loss

We evaluated SARA's autocorrelation-based structural loss Chen et al. (2025a), which encourages structural coherence in generated images. Table 10 shows results at 400K iterations with different loss weights. The adversarial component of SARA caused training instability and was excluded.

Table 10: SARA structural loss ablation at 400K iterations. All experiments use REG + INVAE + SPRINT + RMSNorm + RoPE + QK Norm + Value Residual. The structural loss does not improve upon the baseline.

| Method | FID↓ | sFID↓ | IS↑ | Prec.↑ | Rec.↑ | KDD↓ |
|---|---|---|---|---|---|---|
| Baseline | **3.64** | **5.98** | **202.0** | **0.798** | 0.560 | 0.353 |
| + SARA ($\lambda = 0.5$) | 3.71 | 6.07 | 200.1 | 0.793 | 0.553 | **0.351** |
| + SARA ($\lambda = 0.25$) | 3.80 | 6.00 | 195.6 | 0.786 | **0.563** | 0.361 |

At the default weight ($\lambda = 0.5$), most metrics degraded slightly despite a marginal KDD improvement. Reducing the weight to $\lambda = 0.25$ further degraded performance across most metrics.

### B.6   Alternative Training Objectives

We evaluated alternative training objectives beyond standard flow matching to determine if they could improve generation quality or training efficiency.

**Time-Weighted Contrastive Flow Matching (TCFM).** We hypothesized that CFM's contrastive loss might be detrimental at low noise levels, where it could perturb the learned flow even when the clean image structure is already well-defined. We proposed Time-Weighted CFM (TCFM), which reduces the CFM influence as samples approach the clean image:

$$\mathcal{L}_{\text{TCFM}} = -t \cdot \lambda \mathbb{E}\left[\|v_\theta(x_t, t) - \hat{v}_{\text{target}}\|^2\right] \tag{12}$$

where $t \in [0, 1]$ is the noise timestep. To compensate for the average weighting being halved, we increased $\lambda$ from 0.05 to 0.10. Table 11 shows results comparing CFM and TCFM.

Despite our hypothesis, TCFM underperformed standard CFM across most metrics. The time-weighting did not provide the expected benefit, suggesting that CFM's contrastive signal remains useful even at low noise levels.

$x_0$ **prediction with velocity loss.** We tested the approach from JiT Li & He (2024), which uses $x_0$ prediction combined with velocity-based loss formulation. This technique showed promise for pixel-space

Table 11: CFM vs TCFM ablation at 400K iterations. TCFM does not improve upon standard CFM.

| Method | FID↓ | sFID↓ | IS↑ | Prec.↑ | Rec.↑ | KDD↓ |
|---|---|---|---|---|---|---|
| CFM ($\lambda = 0.05$) | **3.61** | **5.26** | **211.9** | **0.817** | **0.536** | **0.319** |
| TCFM ($\lambda = 0.10$) | 3.82 | 5.57 | 211.8 | 0.813 | 0.533 | 0.321 |

diffusion models. However, in our latent space setting with INVAE, performance was significantly worse than standard flow matching. This suggests that $x_0$ prediction objectives may be most beneficial for pixel-space models rather than latent diffusion.

**Equilibrium Matching.** We attempted to implement Equilibrium Matching Wang & Du (2024), which combines energy-based modeling with flow matching. Despite following the methodology, we were unable to reproduce their reported results on our SiT-B/1 architecture with representation alignment. Performance was significantly worse than standard flow matching, suggesting potential incompatibilities between EqM and our architectural choices or training setup.

## B.7 Alternative Optimizers

We evaluated alternative optimizers to determine if they could improve upon Adam's performance. Specifically, we tested Prodigy Mishchenko & Defazio (2023), which provides adaptive learning rate scheduling.

Table 12: Optimizer comparison on SR-DiT$_4$ (REG + INVAE + SPRINT + RMSNorm + RoPE) at 400K iterations. Adam outperforms Prodigy.

| Optimizer | FID↓ | sFID↓ | IS↑ | Prec.↑ | Rec.↑ | KDD↓ |
|---|---|---|---|---|---|---|
| Adam | **4.09** | 6.22 | **194.2** | 0.778 | **0.563** | 0.367 |
| Prodigy | 4.67 | **5.70** | 175.1 | **0.820** | 0.495 | 0.367 |

On SR-DiT$_4$, Prodigy underperformed Adam (Table 12).

## Visual Results

All class-conditional samples below are generated with 250 sampling steps using path-drop guidance with guidance scale 2.5, guidance low threshold 0.10, and guidance high threshold 0.80.

**Class 68**

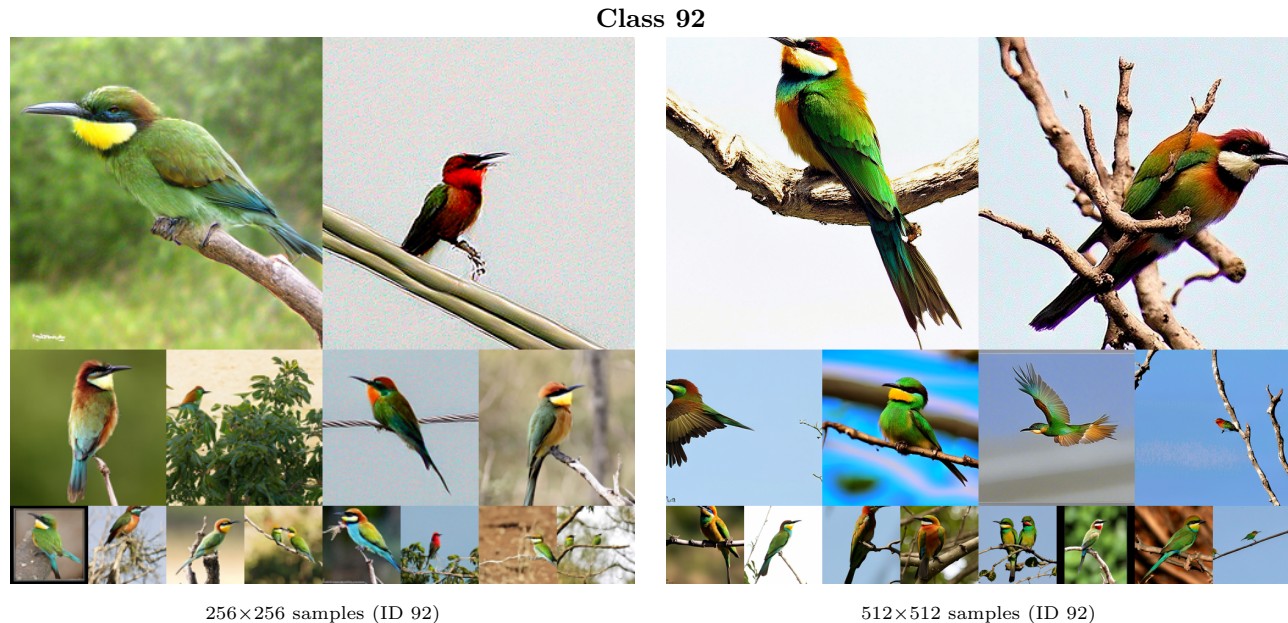

256×256 samples (ID 68)     512×512 samples (ID 68)

Figure 4: Uncurated SR-DiT-B/1 samples for ImageNet class 68 (sidewinder, horned rattlesnake, Crotalus cerastes) at 256×256 and 512×512 resolution.

**Class 92**

256×256 samples (ID 92)     512×512 samples (ID 92)

Figure 5: Uncurated SR-DiT-B/1 samples for ImageNet class 92 (bee eater) at 256×256 and 512×512 resolution.

**Class 233**

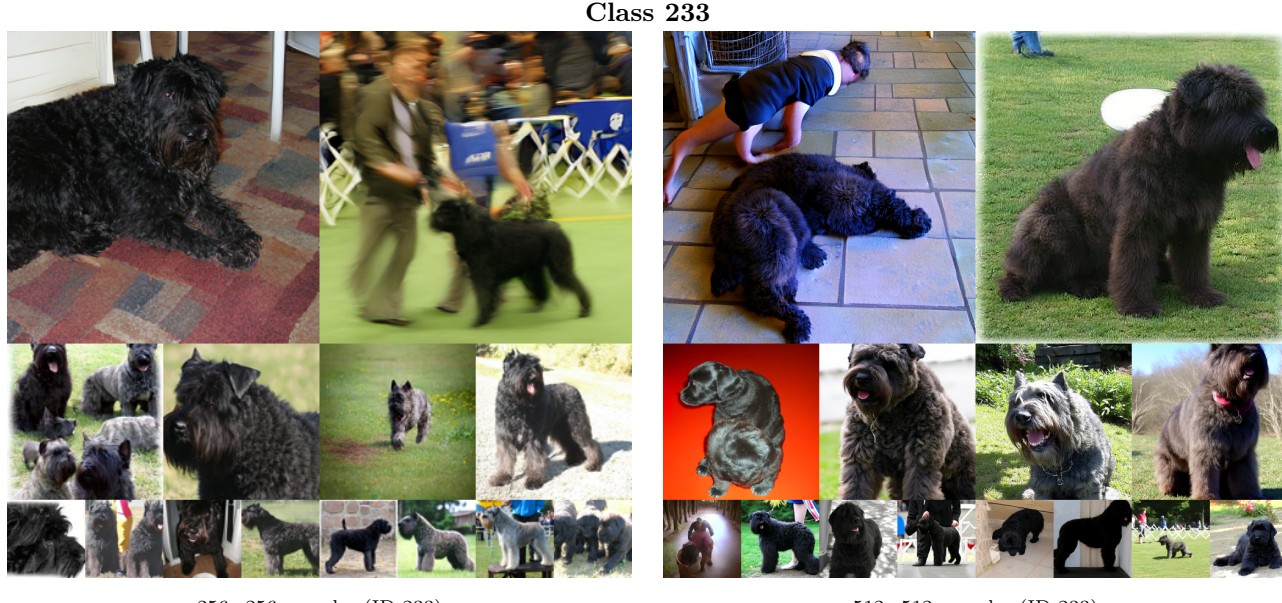

256×256 samples (ID 233)          512×512 samples (ID 233)

Figure 6: Uncurated SR-DiT-B/1 samples for ImageNet class 233 (Bouvier des Flandres, Bouviers des Flandres) at 256×256 and 512×512 resolution.

**Class 273**

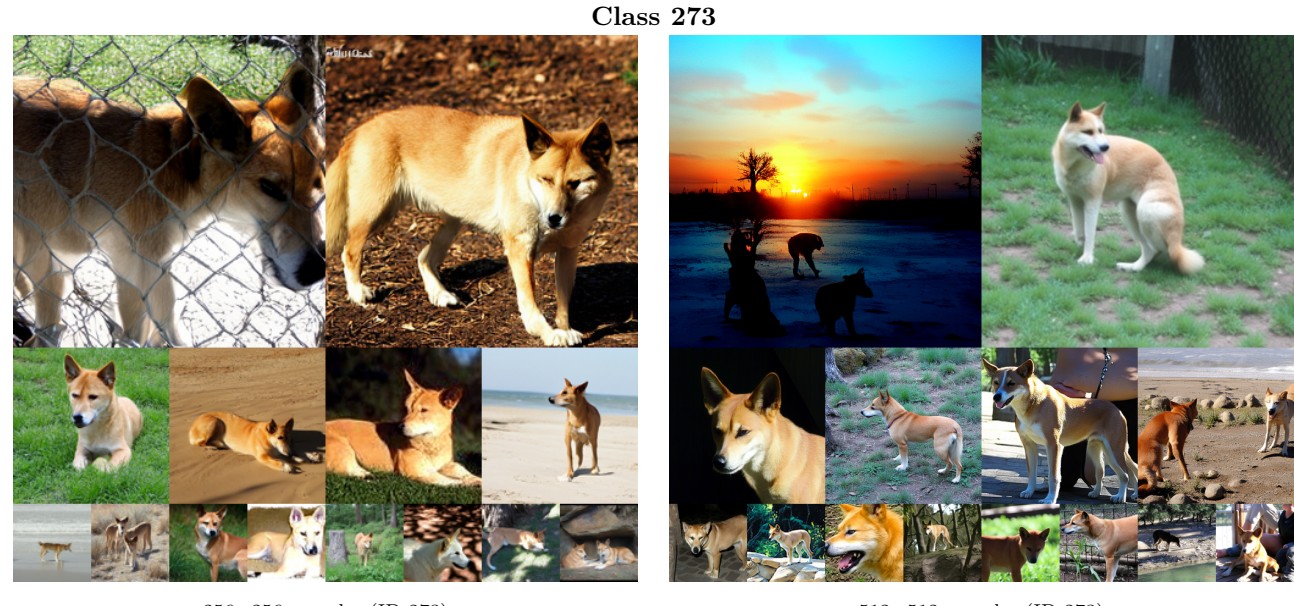

256×256 samples (ID 273)          512×512 samples (ID 273)

Figure 7: Uncurated SR-DiT-B/1 samples for ImageNet class 273 (dingo, warrigal, warragal, Canis dingo) at 256×256 and 512×512 resolution.

**Class 283**

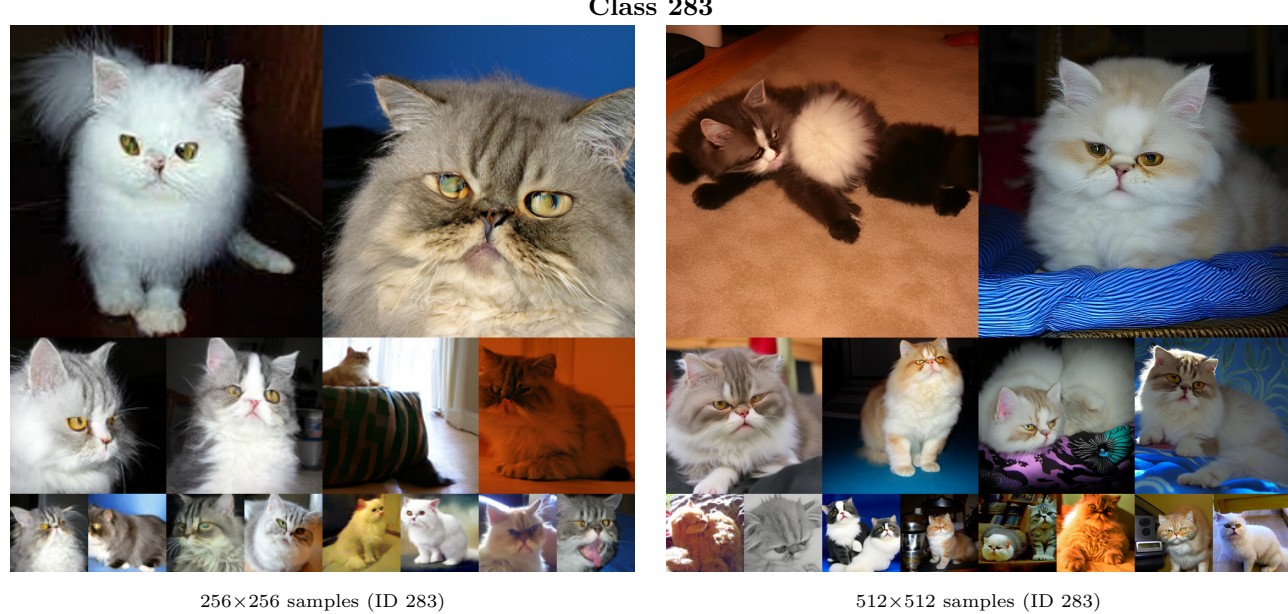

256×256 samples (ID 283)          512×512 samples (ID 283)

Figure 8: Uncurated SR-DiT-B/1 samples for ImageNet class 283 (Persian cat) at 256×256 and 512×512 resolution.

**Class 360**

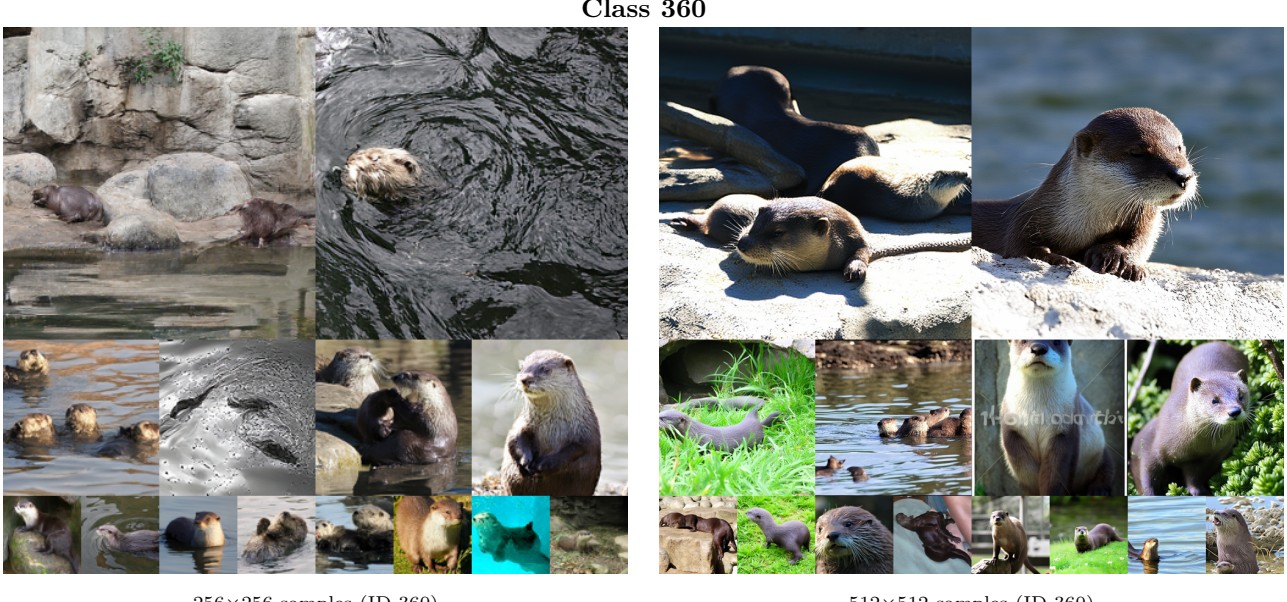

256×256 samples (ID 360)          512×512 samples (ID 360)

Figure 9: Uncurated SR-DiT-B/1 samples for ImageNet class 360 (otter) at 256×256 and 512×512 resolution.

**Class 482**

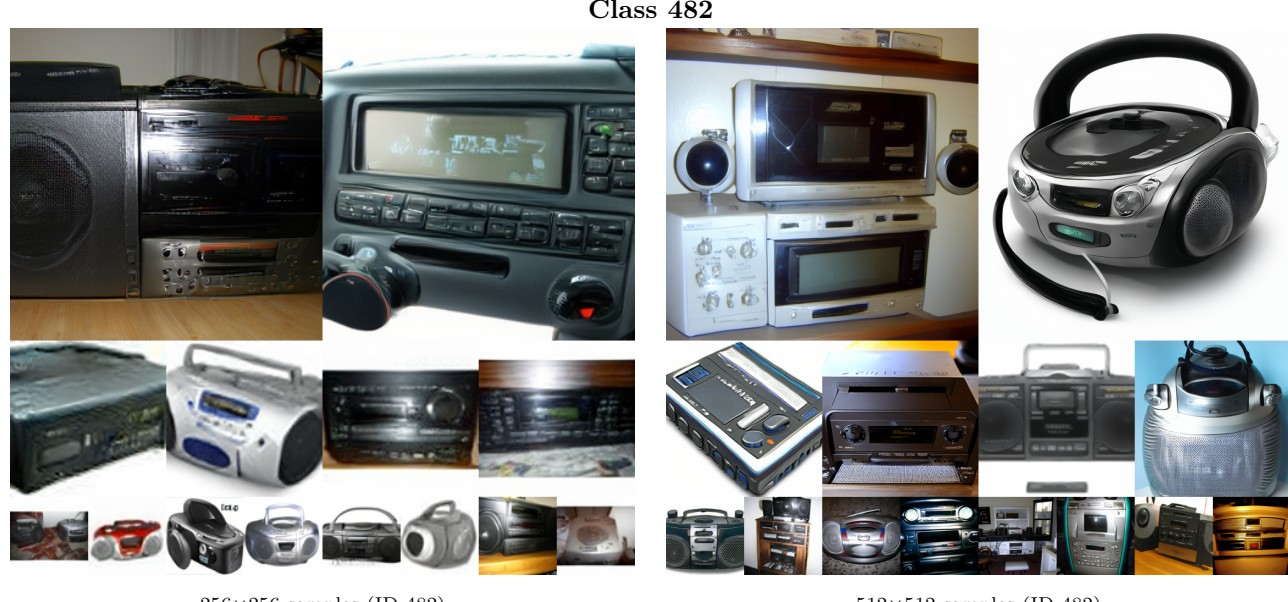

256×256 samples (ID 482)                    512×512 samples (ID 482)

Figure 10: Uncurated SR-DiT-B/1 samples for ImageNet class 482 (cassette player) at 256×256 and 512×512 resolution.

**Class 545**

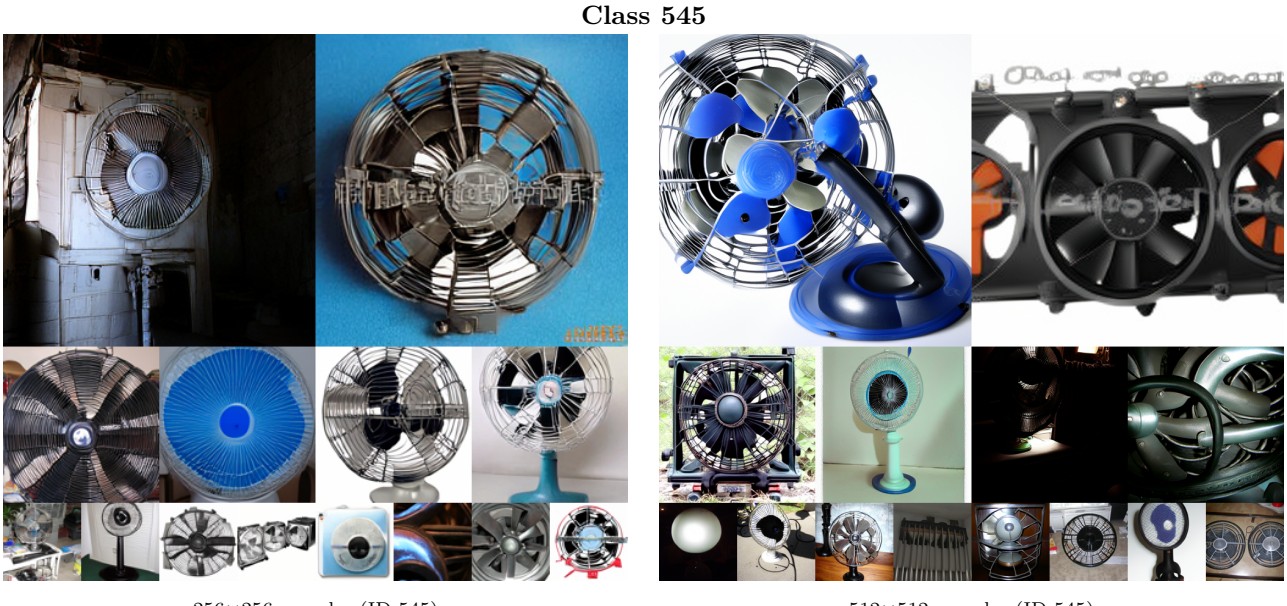

256×256 samples (ID 545)                    512×512 samples (ID 545)

Figure 11: Uncurated SR-DiT-B/1 samples for ImageNet class 545 (electric fan, blower) at 256×256 and 512×512 resolution.

**Class 721**

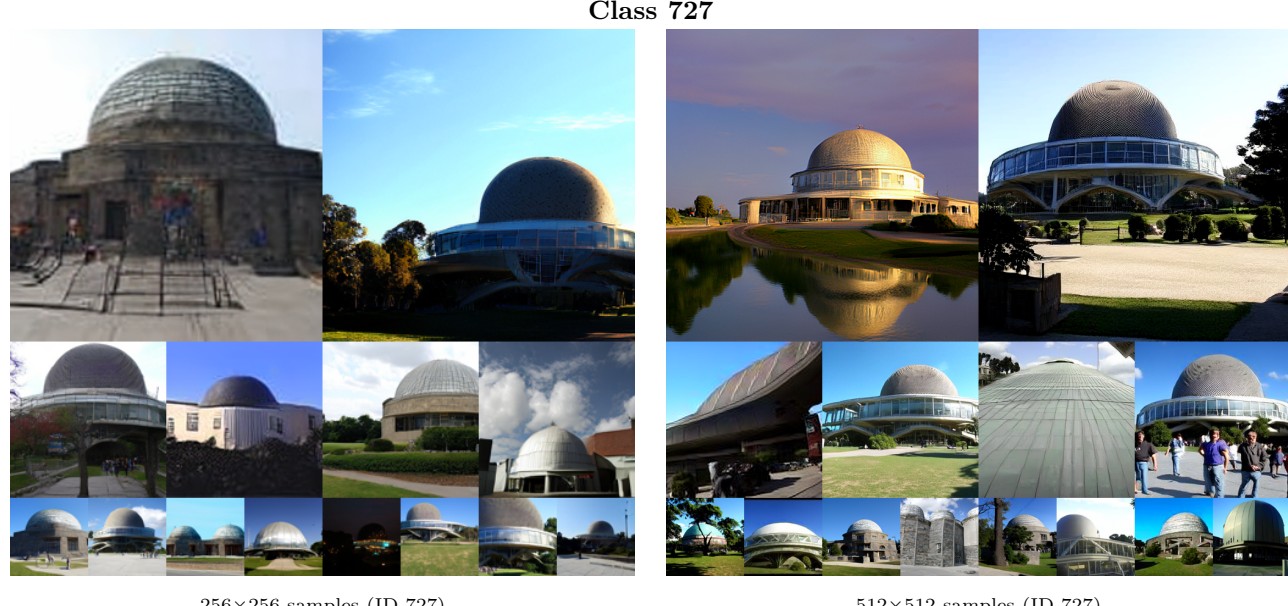

256×256 samples (ID 721)          512×512 samples (ID 721)

Figure 12: Uncurated SR-DiT samples for ImageNet class 721 (pillow) at 256×256 and 512×512 resolution.

**Class 727**

256×256 samples (ID 727)          512×512 samples (ID 727)

Figure 13: Uncurated SR-DiT samples for ImageNet class 727 (planetarium) at 256×256 and 512×512 resolution.

**Class 795**

256×256 samples (ID 795)     512×512 samples (ID 795)

Figure 14: Uncurated SR-DiT samples for ImageNet class 795 (ski) at 256×256 and 512×512 resolution.

**Class 839**

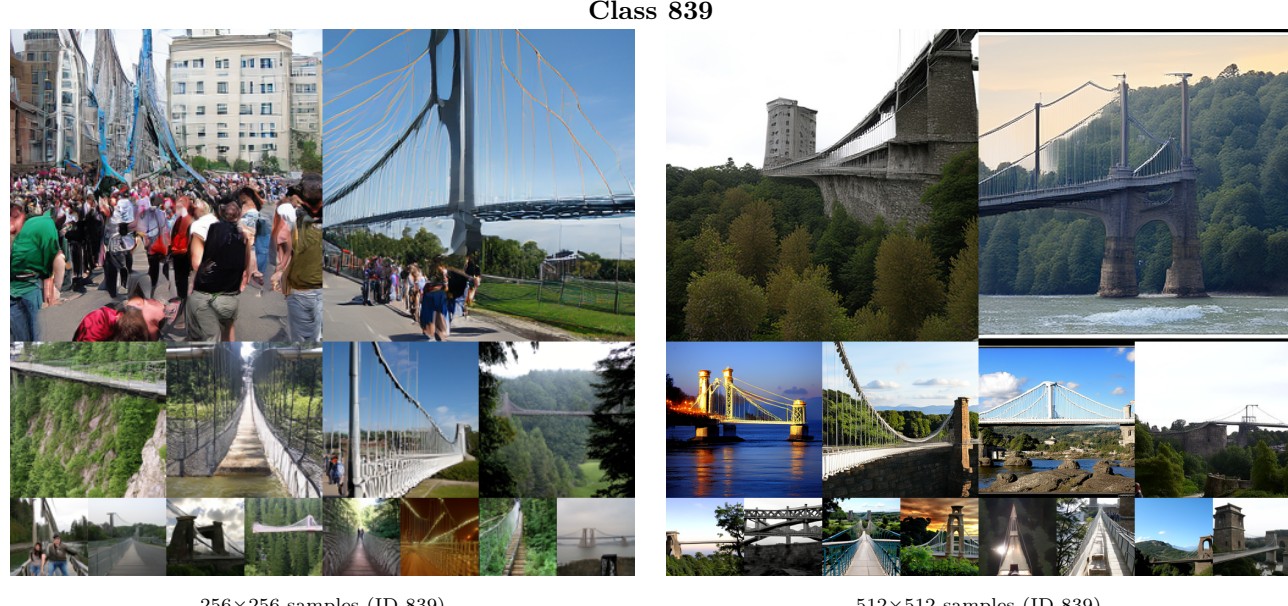

256×256 samples (ID 839)     512×512 samples (ID 839)

Figure 15: Uncurated SR-DiT samples for ImageNet class 839 (suspension bridge) at 256×256 and 512×512 resolution.

**Class 863**

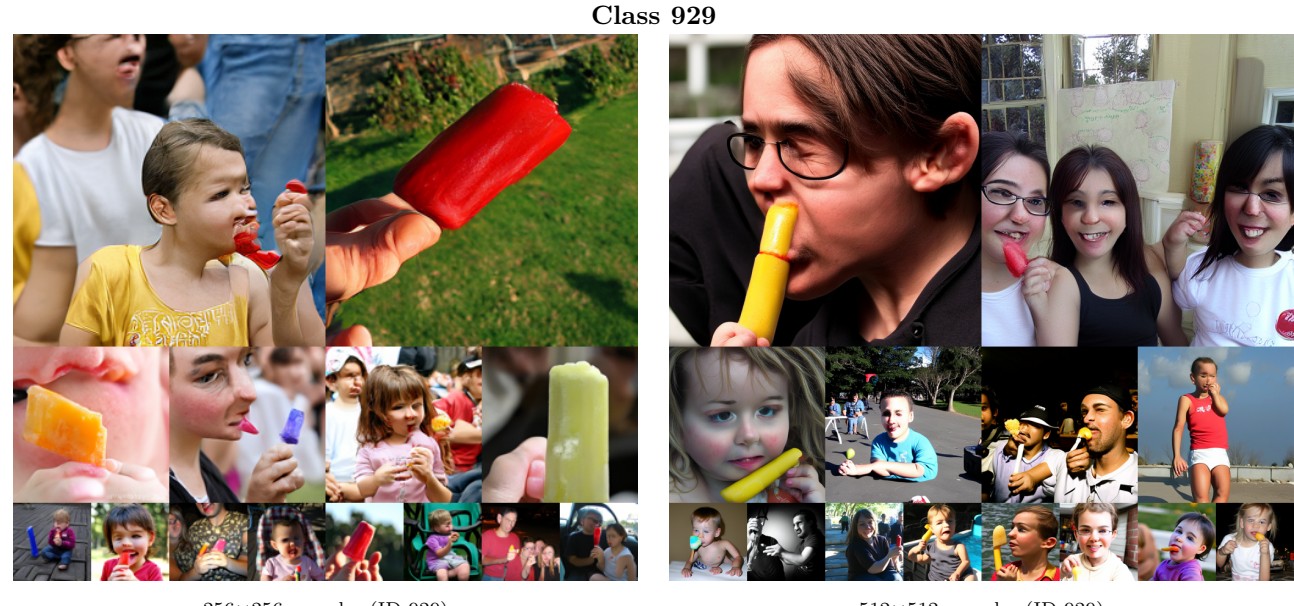

256×256 samples (ID 863)          512×512 samples (ID 863)

Figure 16: Uncurated SR-DiT samples for ImageNet class 863 (totem pole) at 256×256 and 512×512 resolution.

**Class 929**

256×256 samples (ID 929)          512×512 samples (ID 929)

Figure 17: Uncurated SR-DiT samples for ImageNet class 929 (ice lolly, lolly, lollipop, popsicle) at 256×256 and 512×512 resolution.

**Class 944**

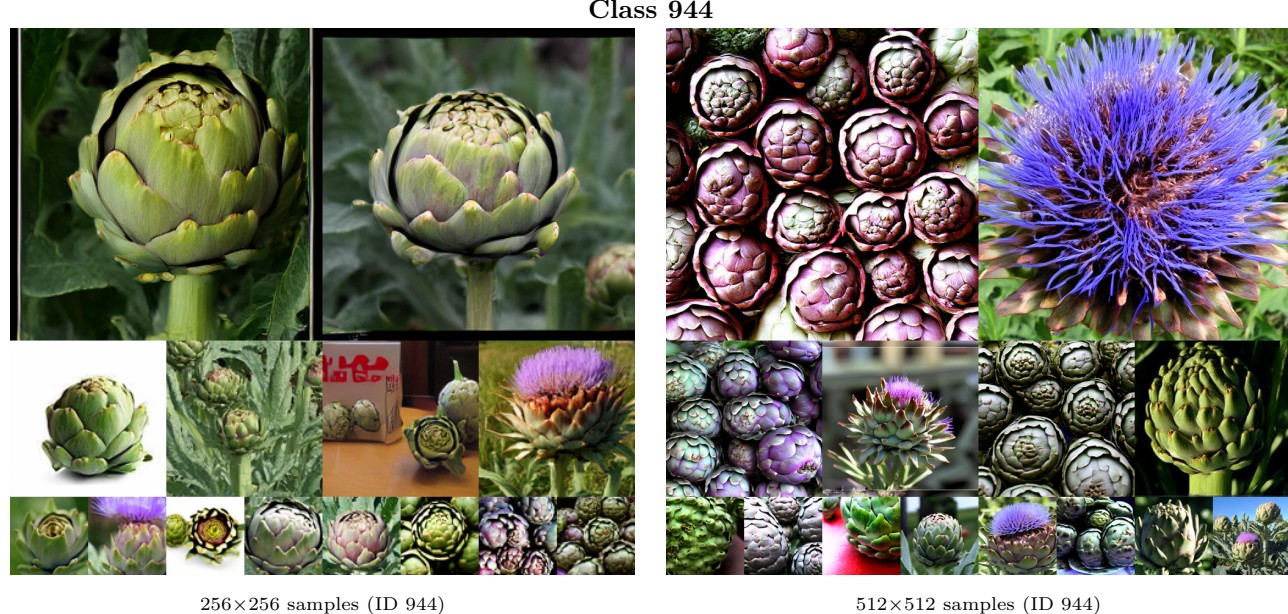

256×256 samples (ID 944)                    512×512 samples (ID 944)

Figure 18: Uncurated SR-DiT samples for ImageNet class 944 (artichoke, globe artichoke) at 256×256 and 512×512 resolution.

**Class 974**

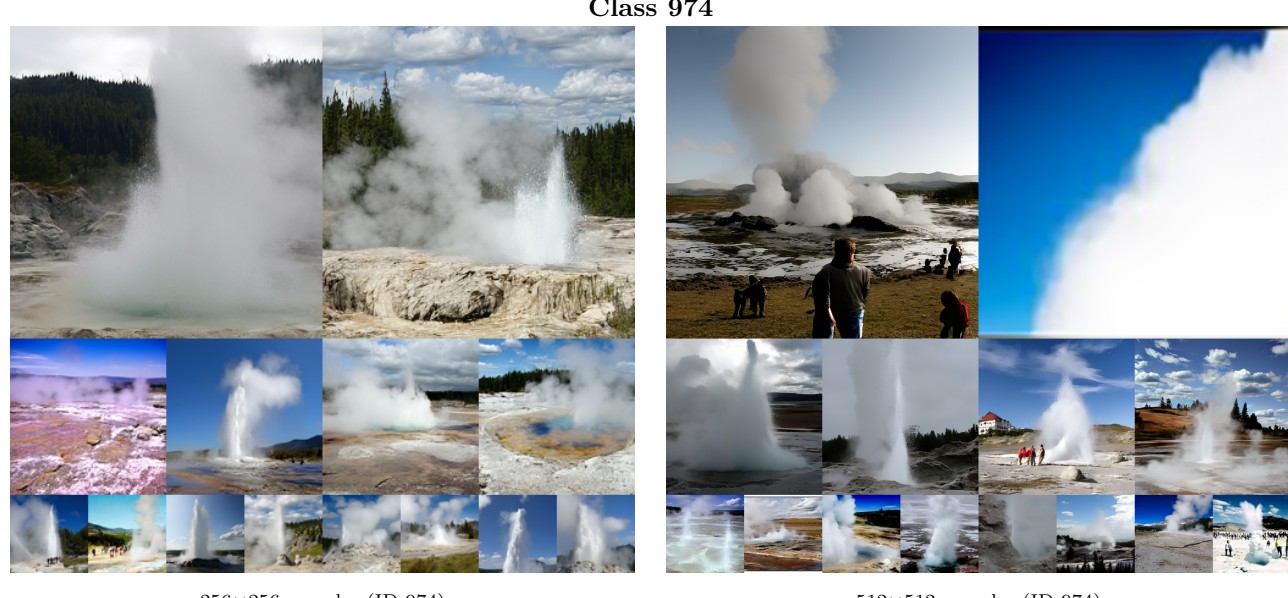

256×256 samples (ID 974)                    512×512 samples (ID 974)

Figure 19: Uncurated SR-DiT samples for ImageNet class 974 (geyser) at 256×256 and 512×512 resolution.

**Class 984**

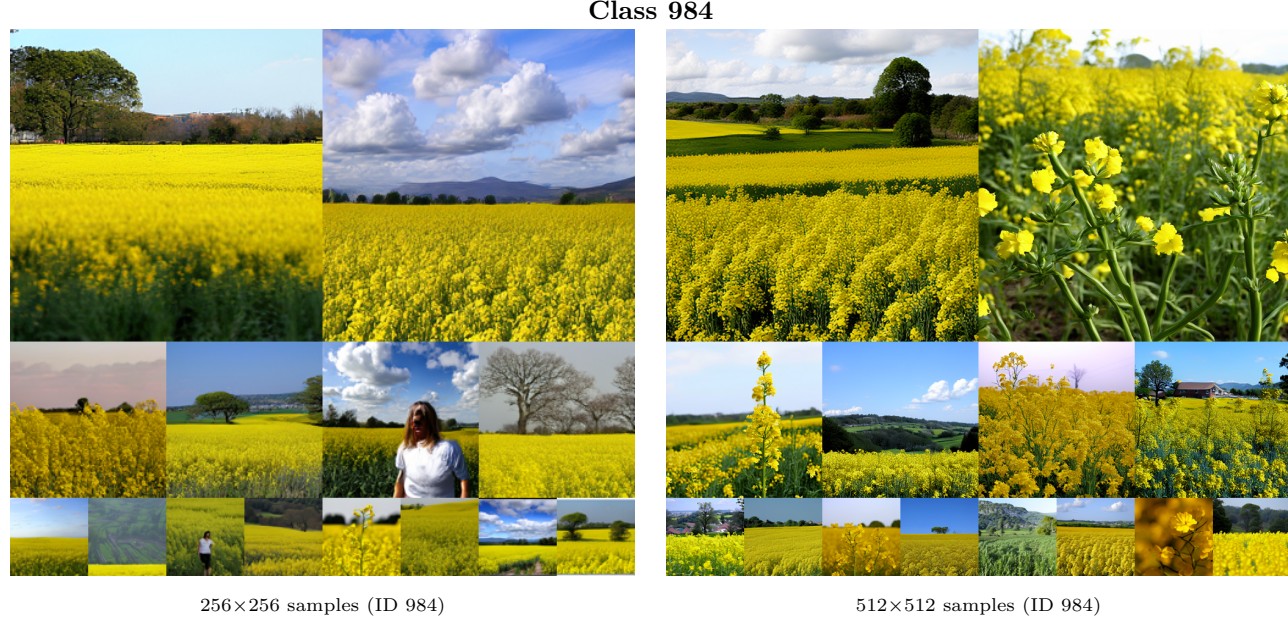

256×256 samples (ID 984)                    512×512 samples (ID 984)

Figure 20: Uncurated SR-DiT samples for ImageNet class 984 (rapeseed) at 256×256 and 512×512 resolution.

**Class 991**

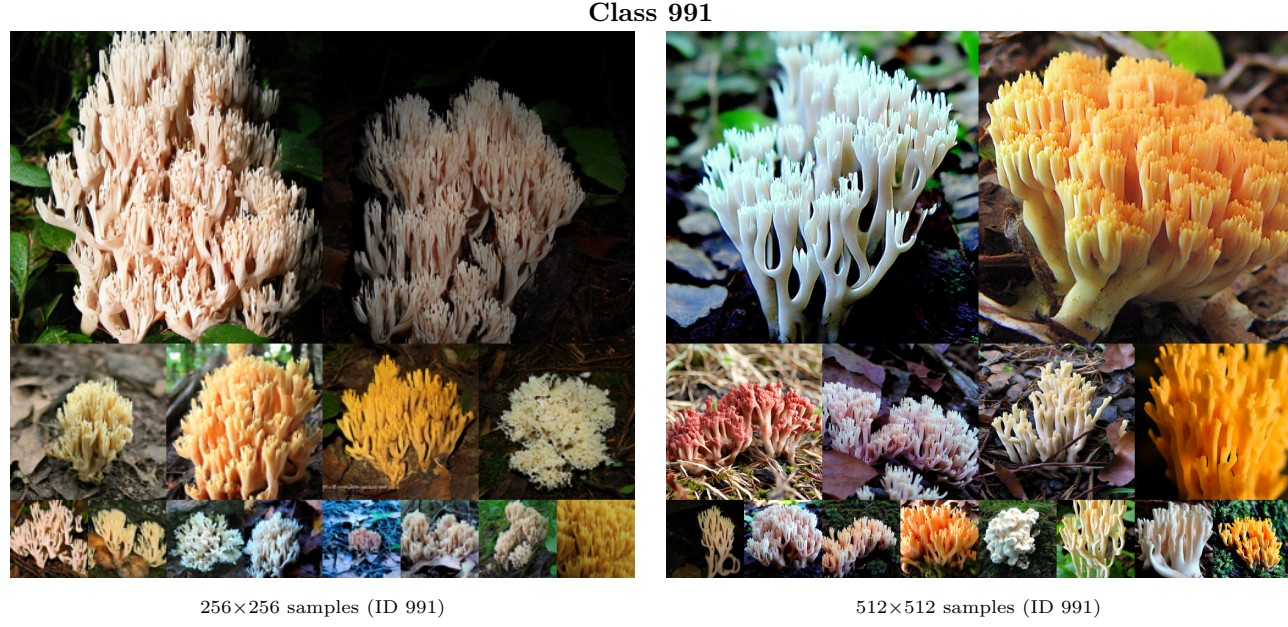

256×256 samples (ID 991)                    512×512 samples (ID 991)

Figure 21: Uncurated SR-DiT samples for ImageNet class 991 (coral fungus) at 256×256 and 512×512 resolution.

