# OpenReview forum: "Speedrunning ImageNet Diffusion"
_TMLR — Under review for TMLR_

### Review · Reviewer_KJur · 2026-05-10

**Summary Of Contributions:**

This paper is mainly an empirical combination study for diffusion transformers. The authors do not introduce one completely new diffusion architecture from scratch. Instead, they take several recent ideas that already existed and test how well they work when used together in one model. The biggest idea is that different methods solve different problems. Some methods help the model learn better semantic features. Some methods reduce wasted computation. Some methods make optimization more stable. The paper argues that if you combine the right ones, the gains stack up. So paper's contribution is not like we invented new ideas but it is more like: we found a strong recipe for training diffusion transformers more efficiently, and we showed which ingredients matter most.

**Audience:**

Yes

**Audience Explanation:**

Intermediate to advanced diffusion-model (DiT) readers, especially people exploring how to improve DiT-style training with modern tricks and ablations.

**Broader Impact Concerns:**

In General for any DiT models, a Broader Impact Statement should explicitly note that efficiency gains are not only beneficial for researchers but can also make harmful uses easier to scale. If such a statement is already present, it should more directly address these risks and explain how the authors think about responsible use, benchmark bias, and potential downstream misuse of the released method or checkpoints.

**Claims And Evidence:**

Yes

**Claims Explanation:**

Mostly yes, but with caveats. The submission provides substantial empirical evidence through head-to-head comparisons, ablations, and convergence results, and the main claims are plausible and fairly well supported within the reported experimental setting. However, the evidence is not conclusive enough to establish broad generality beyond ImageNet and the specific SR-DiT/REG/IN-VAE configuration studied.

**Requested Changes:**

The submission already reports a solid and appropriate set of metrics for DiT evaluation, including FID, KDD, sFID, IS, precision, and recall. I do not think additional metrics are necessary. However, the ablation story would be stronger if the authors tested the same combinations on at least one additional dataset, resolution, or training budget, to show that the observed gains are not specific to a single ImageNet setting.
The paper’s efficiency claim is plausible, but the reason larger baselines underperform is not fully isolated. It is unclear whether the gap comes from model size itself, insufficient training budget, different optimization sensitivity, or the interaction between the recipe and scale. A controlled scaling study would make this claim much more convincing.

---

> ### Author Response · Authors · 2026-06-22
>
> > **Generality — testing the same combinations on at least one additional dataset, resolution, or training budget would show the gains are not specific to a single ImageNet setting.**
>
> We are unable to add new datasets in this revision, but we now make explicit that our existing results already provide partial evidence along two of the three axes the reviewer names: the recipe is evaluated at two resolutions (256 and 512) and across multiple training budgets (intermediate checkpoints at 50K/100K/200K/400K iterations), with consistent gains in both. This is stated in the new "Generality across settings" limitation, together with an explicit acknowledgment that broadening beyond ImageNet remains important future work.
>
> > **Efficiency claim — the reason larger baselines underperform is not isolated (model size vs. training budget vs. optimization sensitivity vs. recipe×scale interaction). A controlled scaling study would be more convincing.**
>
> We agree that the source of the gap is worth isolating, and we address this in three parts. (i) We do not claim that a 140M model is intrinsically superior to a larger one; our claim is state-of-the-art *at this model size* (abstract). (ii) The baselines underperform at matched iterations primarily because they lack most of the recipe — a semantic VAE, token routing, and the architecture/optimizer stack; even the strong REPA/REG baselines share only representation alignment with SR-DiT — rather than because larger capacity is harmful. (iii) We agree that our comparison does not cleanly separate the contribution of model size from those of training budget, optimizer sensitivity, and the recipe×scale interaction. A controlled scaling study — the identical recipe applied across SiT-B/L/XL — is the natural way to disentangle these; it was beyond our compute budget for this revision, and we identify it as the most valuable direction for future work. The revised Limitations notes this isolation gap explicitly as a limitation.
>
> > **Broader Impact — the statement should note that efficiency gains can make harmful uses easier to scale, and address responsible use, benchmark bias, and downstream misuse.**
>
> We have added a Broader Impact Statement covering: (i) the dual-use nature of training-efficiency gains — the same speedups that broaden access for academic groups also lower the barrier for harmful uses such as disinformation and non-consensual or impersonating imagery, and allow them to be produced at larger scale; (ii) the scope of our released artifacts — the checkpoints are class-conditional ImageNet generators over 1000 fixed categories (not conditioned on text, faces, or individuals), which limits their direct potential for targeted misuse, while we note that the recipe is general and recommend standard safeguards (content provenance and watermarking, dataset documentation, output filtering, and gated or staged release) for anyone scaling it to more capable settings; and (iii) benchmark and metric bias — ImageNet's known demographic, geographic, and category-level biases and the feature-extractor biases inherited by FID/IS/KDD, with the caveat that improvements on these aggregate metrics do not imply fairness or balanced coverage and should be paired with task-appropriate auditing.

---

### Review · Reviewer_aCcA · 2026-05-18

**Summary Of Contributions:**

This paper proposes SR-DiT (Speedrun Diffusion Transformer), a framework that systematically combines recent advances in efficient diffusion transformer training, including representation alignment (REG/REPA), semantic latent spaces (INVAE), token routing (SPRINT), architectural modifications (RMSNorm, RoPE, QK normalization, Value Residual Learning), and training improvements such as Contrastive Flow Matching and time shifting. The method achieves strong ImageNet-256 performance with FID 3.14 and KDD 0.290 using only a 140M parameter model trained for 400K iterations. The paper is well written, experimentally thorough, and practically relevant to the diffusion modeling community. The extensive ablation studies and inclusion of negative results are valuable. However, several aspects require clarification and stronger validation.

**Audience:**

Yes

**Audience Explanation:**

The paper can be used as a good baseline for future improvements in diffusion models.

**Broader Impact Concerns:**

No concerns

**Claims And Evidence:**

Yes

**Claims Explanation:**

Table 1 reports the improvements in the metrics.

**Requested Changes:**

Major Comments

1.	The paper evaluates improvements through a single progressive integration path, where techniques are added sequentially on top of earlier modifications. However, it remains unclear whether the observed importance of individual components is dependent on this specific ordering. A discussion or limited study of ordering effects and interaction dependencies between components would strengthen the paper’s claims regarding which techniques matter most.

2.	The primary contribution appears to be the integration of multiple existing techniques rather than the introduction of a fundamentally new algorithmic idea. While the empirical performance is good, the paper should articulate the conceptual insights gained from combining these components beyond engineering optimization.

3.	The paper emphasizes FID improvements heavily, but some metrics indicate tradeoffs between precision and recall. Later SR-DiT variants improve precision while sometimes reducing recall. A more detailed discussion of the diversity-quality tradeoff would strengthen the evaluation.

4.	The ImageNet-512 comparison is relatively limited and includes only a small number of baselines. Additional comparisons against recent efficient diffusion transformer methods trained under similar compute budgets would strengthen the state-of-the-art claims.

Minor Comments / Suggestions

1.	The paper introduces many abbreviations and shorthand method names in rapid succession, which can make the paper difficult to follow, especially for readers not already deeply familiar with recent diffusion literature. Consider adding a concise terminology/acronym table or brief reminder description.

---

> ### Author Response · Authors · 2026-06-22
>
> > **Major #1 — The single progressive integration path leaves it unclear whether component importance depends on the specific ordering.**
>
> We have added a discussion, "On the ordering of components," that separates two questions. First, the *final* configuration is order-invariant: the full set of components reaches the same model regardless of the order in which they are added. Second, the *per-component* marginal contributions are not order-independent — a component's measured value depends on which others are already present. We make this concrete with RELU² (Appendix): the same swap from GELU to RELU² *improves* FID on its own (4.02 → 3.81) but *worsens* it when Value Residual is already present (3.64 → 3.81). The two interact antagonistically — Value Residual's 0.38-FID gain on GELU is cancelled on RELU² — so the same change helps or hurts depending only on order. We are therefore explicit that fully characterizing such interactions would require evaluating each component against many partial subsets, an O(n²) study rather than the O(n) of a single progressive path, which was beyond our compute budget; we present the per-component deltas as path-dependent attributions rather than context-free contributions, and note residual order-sensitivity as an explicit limitation.
>
> > **Major #2 — The paper should articulate the conceptual insights gained beyond engineering optimization.**
>
> We have made the conceptual content explicit through the new "A taxonomy of training bottlenecks" paragraph and table and the "Synergy vs. redundancy" paragraph (also described in our response to Reviewer v4Tg's W1). We organize the techniques by the training bottleneck each targets, quantify the relative FID reduction attributable to each axis, and relate the same view to our negative results. We deliberately frame this as a descriptive organizing lens and a retrospective account of our results, rather than a tested predictive principle, to avoid over-claiming.
>
> > **Major #3 — Some metrics show precision/recall tradeoffs; the diversity-quality tradeoff deserves more discussion.**
>
> We have added a "Quality–diversity trade-off" paragraph making the precision/recall behavior explicit. Across the progression, precision rises substantially (0.691 → 0.807) while recall declines from 0.632 to a low of 0.536 before partially recovering to 0.565; the two largest recall costs coincide with SPRINT and CFM (the latter expected, since CFM's contrastive term deliberately sharpens samples). We treat this as a modest but real diversity cost rather than emphasizing the FID gains alone. KDD — a kernel distance between the generated and real feature distributions, and hence sensitive to coverage as well as quality — nonetheless decreases overall (0.586 → 0.290). We also make the tradeoff actionable: because the recall cost is concentrated in SPRINT and CFM, an application that prioritizes diversity over peak FID or efficiency can omit those two components, trading fidelity and efficiency for coverage.
>
> > **Major #4 — The ImageNet-512 comparison is limited to a small number of baselines.**
>
> We have added a caveat after the ImageNet-512 table. Most recent efficient diffusion transformers report ImageNet-512 numbers only with classifier-free guidance, with much longer training, or with larger models, so few results are directly comparable to our setting (no guidance, 400K iterations, B-sized model) and the comparable baseline set is correspondingly small. We now frame the 512 results as evidence that the recipe transfers to higher resolution rather than as an exhaustive state-of-the-art benchmark, and we note that SR-DiT-B/1 is in fact smaller than the DiT-XL/2 baseline it surpasses. Broadening the set of matched-compute ImageNet-512 baselines is a useful direction for future evaluation.
>
> > **Minor — Many abbreviations and shorthand method names are introduced quickly; consider a terminology/acronym table.**
>
> We have added a terminology and abbreviations table immediately after the contributions, defining each shorthand name and metric used throughout — SR-DiT, SiT, REPA, REG, INVAE, SPRINT, TREAD, CFM, Value Residual, RoPE, RMSNorm, QK Norm, Muon, CFG, PDG, NFE, KDD, FID, sFID, IS, Precision/Recall, DINOv2, and SD-VAE — each with a one-line description and citation, so that readers can refer back at any point.

---

### Review · Reviewer_v4Tg · 2026-06-14

**Summary Of Contributions:**

This paper presents SR-DiT (Speedrun Diffusion Transformer), a framework for improving the training efficiency of diffusion transformers by systematically combining several recently proposed techniques.

The proposed framework builds upon a stronger semantic VAE (INVAE), advanced representation alignment methods (REPA/REG), and token routing (SPRINT). In addition, the authors integrate a collection of modern architectural modifications, including RMSNorm, RoPE, QK normalization, and Value Residual Learning, together with several training improvements such as Contrastive Flow Matching, Time Shifting, Balanced Sampling, Multi-Layer REPA, Muon, and Layerwise Scaling.

The resulting model, SR-DiT-B/1, achieves FID 3.14 and KDD 0.290 on ImageNet-256 at 400K training iterations using only 140M parameters and without classifier-free guidance. The paper further provides extensive ablation studies that quantify the contribution of each component and investigates both positive interactions and incompatibilities among techniques.

Strengths:
1. The paper provides a relatively systematic study of the effectiveness of recent advanced DiT-specific and general-purpose training techniques for diffusion transformer training.
2. The paper explores the performance boundary of DiT on the ImageNet-1K generation task under current training techniques.
3. The paper provides a computationally efficient baseline that achieves state-of-the-art performance with relatively modest compute resources.

Weaknesses
1. The primary contribution of the paper lies in the systematic integration and evaluation of existing techniques. While this is not a major concern under the TMLR acceptance criteria.
2. The experimental validation is limited to relatively small-scale ImageNet class-conditional generation. The paper lacks comparisons and validation on larger models (e.g., DiT-XL scale), longer training schedules, more challenging evaluation settings, and more general generation tasks such as text-to-image generation. Some of these limitations are acknowledged by the authors in Section 6.1.

**Audience:**

Yes

**Audience Explanation:**

I believe the findings would be of interest to at least a subset of the TMLR audience, particularly:

Researchers studying the training efficiency of diffusion transformers.

Researchers interested in understanding the effectiveness and compatibility of recent DiT training techniques.

Engineers and practitioners developing advanced image generation models, who may benefit from the proposed computationally efficient training recipe and baseline.

**Broader Impact Concerns:**

I do not have any specific ethical concerns regarding this work.

**Claims And Evidence:**

Yes

**Claims Explanation:**

The paper explores the effectiveness of combining a diverse set of recent advanced training techniques for diffusion transformers. The main claim, that the integration of these techniques can substantially improve DiT training efficiency, is supported by comprehensive ablation studies on ImageNet-256.

**Requested Changes:**

My main concern with the paper is reflected in Weakness 2. Nevertheless, I view this primarily as a limitation of the experimental scope rather than a fundamental issue with the paper. Therefore, it does not significantly affect my overall positive assessment, and I remain inclined to support acceptance.

---

> ### Author Response · Authors · 2026-06-22
>
> > **W1 — The primary contribution is the systematic integration and evaluation of existing techniques.**
>
> We agree, and we have made the conceptual content of that integration explicit. The revised Analysis adds a paragraph, "A taxonomy of training bottlenecks," and an accompanying table that organize the techniques by the bottleneck each primarily targets — semantic learning signal; computational redundancy and information flow; expressivity and stability; optimization; and loss/sampling shaping — annotating each axis with the relative FID reduction it contributes. We present this explicitly as a descriptive organizing lens, not a tested claim that the categories are independent. A companion paragraph, "Synergy vs. redundancy," observes that several of our negative results (dispersive loss, SARA, and Time-Weighted CFM) share a pattern — a technique tends not to help when its target bottleneck is already addressed — which we offer as a post-hoc account of what we observed rather than a validated rule. We thank the reviewer for noting that the integrative nature of the work is not a major concern under the TMLR criteria.
>
> > **W2 — Validation is limited to small-scale ImageNet class-conditional generation (no DiT-XL scale, longer training, or text-to-image).**
>
> Our efficiency claim is already scoped to model size — the abstract reports a state-of-the-art result *at this model size*, and we do not claim that a small model is intrinsically superior to a larger one. To address the reviewer's concern directly, the revised Limitations adds a "Scope of the efficiency claim" item noting that, because our comparison is against larger models that lack our recipe, it demonstrates the recipe's value at a fixed compute budget but does not isolate the effect of model size from those of the recipe and training budget; a controlled scaling study (the same recipe across SiT-B/L/XL) is the natural way to disentangle these and is identified as future work. We have also added a "Generality across settings" limitation, noting that our existing results already span two resolutions (256 and 512) and the full training trajectory, while agreeing that additional datasets and text-to-image generation remain important future directions. We appreciate the reviewer's view that these are limitations of experimental scope rather than fundamental issues.

---

### Author Response · Authors · 2026-06-22

We thank all three reviewers for their careful and constructive reviews, and for their positive overall assessments. We have revised the paper in response to the comments. The revision is text-only — clarifications and additional analysis of our existing results rather than new experiments; comments that call for experiments beyond our current compute (a scaling study, additional datasets) are acknowledged as limitations. We summarize the main additions here before responding to each point individually.

Summary of changes:

1. A new Broader Impact Statement addressing the dual-use nature of efficiency gains, the scope of the released checkpoints, and benchmark/metric bias.
2. A terminology and abbreviations table near the start of the paper, defining every shorthand method name and metric.
3. A taxonomy of training bottlenecks (a new paragraph and table) organizing the integrated techniques by the bottleneck each primarily targets, with the relative FID reduction attributable to each axis, together with a "Synergy vs. redundancy" paragraph relating this view to our negative results.
4. A "Quality–diversity trade-off" paragraph making the precision/recall behavior explicit, with practical guidance.
5. An "On the ordering of components" discussion (and a matching limitation) analyzing how per-component contributions depend on the integration order.
6. A "Scope of the efficiency claim" limitation clarifying what our efficiency result does and does not establish.
7. A caveat on the ImageNet-512 comparison and an expanded Limitations section on generality across settings.